# SKOLR: Structured Koopman Operator Linear RNN for Time-Series Forecasting

Yitian Zhang [1 2 3]  Liheng Ma [1 2 3]  Antonios Valkanas [1 2 3]  Boris N. Oreshkin [4]  Mark Coates [1 2 3]

## Abstract

Koopman operator theory provides a framework for nonlinear dynamical system analysis and time-series forecasting by mapping dynamics to a space of real-valued measurement functions, enabling a linear operator representation. Despite the advantage of linearity, the operator is generally infinite-dimensional. Therefore, the objective is to learn measurement functions that yield a tractable finite-dimensional Koopman operator approximation. In this work, we establish a connection between Koopman operator approximation and linear Recurrent Neural Networks (RNNs), which have recently demonstrated remarkable success in sequence modeling. We show that by considering an extended state consisting of lagged observations, we can establish an equivalence between a structured Koopman operator and linear RNN updates. Building on this connection, we present **SKOLR**, which integrates a learnable spectral decomposition of the input signal with a multilayer perceptron (MLP) as the measurement functions and implements a structured Koopman operator via a highly parallel linear RNN stack. Numerical experiments on various forecasting benchmarks and dynamical systems show that this streamlined, Koopman-theory-based design delivers exceptional performance. Our code is available at: https://github.com/networkslab/SKOLR.

## 1. Introduction

Time-series prediction and analysis of nonlinear dynamical systems remain fundamental challenges across various domains. Koopman operator theory (Koopman, 1931) offers a promising mathematical framework that transforms nonlinear dynamics into linear operations in the space of measurement functions. Practical implementation of this theory faces significant challenges due to the infinite dimensionality of the resulting linear operator, necessitating finite dimensional approximations. The dynamic mode decomposition (DMD) and its variants are the most widely employed approximations (Rowley et al., 2009; Schmid, 2010; Williams et al., 2015), although alternative techniques have emerged (Bevanda et al., 2021; Khosravi, 2023), including ones employing learnable neural measurement functions (Li et al., 2017)

In parallel developments, linear Recurrent Neural Networks (linear RNNs) have emerged as a powerful architecture in deep learning and sequence modeling (Stolzenburg et al., 2018; Gu & Dao, 2023; Wang et al., 2024b). These models leverage the computational efficiency of linear recurrence while maintaining impressive modeling capabilities.

In this work, we consider the task of time-series forecasting and establish both an explicit connection and a direct architectural match between Koopman operator approximation and linear RNNs. In particular, we show that by representing the dynamic state using a collection of time-delayed observations, we can establish an equivalence between the application of an extended DMD-style approximation of the Koopman operator and the state update of a linear RNN.

Building on this connection, we introduce **S**tructured **K**oopman **O**perator **L**inear **R**NN (SKOLR) for time-series forecasting. SKOLR implements a structured Koopman operator through a highly parallel linear RNN stack. Through a learnable spectral decomposition of the input signal, the RNN chains jointly attend to different dynamical patterns from different representation subspaces, creating a theoretically-grounded yet computationally efficient design that naturally aligns with Koopman principles.

Through extensive experiments on various forecasting benchmarks and dynamical systems, we demonstrate that

[1]Department of Electrical and Computer Engineering, McGill University, Montreal, Canada [2]Mila - Quebec Artificial Intelligence Institute, Montreal, Canada [3]ILLS - International Laboratory on Learning Systems, Montreal, Canada [4]Amazon Science. This work does not relate to the author's position at Amazon. Correspondence to: Yitian Zhang <yitian.zhang@mail.mcgill.ca>, Liheng Ma <liheng.ma@mail.mcgill.ca>.

*Proceedings of the 42$^{nd}$ International Conference on Machine Learning*, Vancouver, Canada. PMLR 267, 2025. Copyright 2025 by the author(s).

this streamlined, Koopman-theory-based design delivers exceptional performance, while maintaining the simplicity of the linear RNN and its outstanding parameter efficiency.

## 2. Preliminary

This section provides foundational background for the proposed forecasting methodology. We define the discrete-time dynamical systems used to model observed time series and then introduce the Koopman operator.

**Definition 2.1** (Discrete-time Dynamical Systems). We consider the (autonomous) discrete-time dynamical system:

$$\mathbf{x}_{k+1} = F(\mathbf{x}_k) \tag{1}$$

where $\mathbf{x}_k \in \mathcal{M}$ denotes the system state at time $k \in \mathbb{Z}^+$; and $F : \mathcal{M} \to \mathcal{M}$ represents the underlying dynamics mapping the state forward in time. We assume a Euclidean state space $\mathcal{M} \subset \mathbb{R}^C$, although it can be more generally defined on an $n$-dimensional manifold (Bevanda et al., 2021).

The Koopman operator framework enables globally linear representations of nonlinear systems by applying a linear operator to measurement functions (observables) $g$ of the state $\mathbf{x}_k$. The following theorem formalizes key properties of the Koopman operator.

**Theorem 2.2** (Koopman Operator Theorem (Koopman, 1931; Brunton et al., 2022)). *Considering real-valued measurement functions (a.k.a. observables) $g : \mathcal{M} \to \mathbb{R}$, the Koopman operator $\mathcal{K} : \mathcal{F} \to \mathcal{F}$ is an infinite-dimensional linear operator on the space of all possible measurement functions $\mathcal{F}$, which is an infinite-dimensional Hilbert space, satisfying:*

$$\mathcal{K} \circ g = g \circ F, \tag{2}$$

*where $\circ$ is the composition operator.*

*In other words,*

$$\mathcal{K}(g(\mathbf{x}_k)) = g(F(\mathbf{x}_k)) = g(\mathbf{x}_{k+1}). \tag{3}$$

*This is true for any measurement function $g$ and for any state $\mathbf{x}_k$.*[1]

While this facilitates analysis via linear maps, Koopman operator is generally infinite-dimensional, acting on a Hilbert space of functions. For practical learning and inference, we seek effective finite-dimensional approximations. In this paper, we construct these approximations efficiently by leveraging a connection to linear recurrent neural networks (RNNs). For clarity, we now define a linear RNN.

**Definition 2.3** (Linear Recurrent Neural Network (Stolzenburg et al., 2018)). Consider a hidden state space $\mathcal{H} \subseteq \mathbb{R}^{d_h}$

---

[1] Koopman operators can also be defined on a continuous-time dynamical system, which we disregard here for now.

and input space $\mathcal{V} \subseteq \mathbb{R}^{d_v}$. For any sequence $(\mathbf{v}_k)_{k=1}^L \in \mathcal{V}$, the linear RNN defines a discrete-time dynamical system through the hidden state transition equation:

$$\mathbf{h}_k = \mathbf{W}\mathbf{h}_{k-1} + \mathbf{U}\mathbf{v}_k + \mathbf{b} \tag{4}$$

where $\mathbf{W} \in \mathbb{R}^{d_h \times d_h}$ is the hidden state transition matrix, $\mathbf{U} \in \mathbb{R}^{d_h \times d_v}$ is the weight matrix applied to the input, and $\mathbf{b} \in \mathbb{R}^{d_h}$ is the bias vector. The evolution of hidden states $\mathbf{h}_k \in \mathcal{H}$ is uniquely determined by this linear map.

In order to prepare the connection to our proposed Koopman operator learning strategy and forecasting method, let us introduce $\mathbf{v}_k := \psi(\mathbf{y}_k)$ and define $g(\mathbf{y}_k) := \mathbf{U}\psi(\mathbf{y}_k) + \mathbf{b}$ for a suitable function $\psi$. Then, if $g(\mathbf{y}_{k-s}) = 0$ for $s > L$, we can unroll Eq. 4 to the following form:

$$\mathbf{h}_k = g(\mathbf{y}_k) + \sum_{s=1}^{L} \mathbf{W}^s g(\mathbf{y}_{k-s}). \tag{5}$$

Here $\mathbf{W}^s$ indicates $s$ applications of $\mathbf{W}$.

## 3. Methodology

### 3.1. Problem Statement

Let us denote $L$ steps of the trajectory of a discrete-time dynamical system as $\mathbf{x}_1, \ldots, \mathbf{x}_L$. We focus on the setting where $\mathbf{x}_k \in \mathcal{X} \subseteq \mathbb{R}^C$. We do not directly observe $\mathbf{x}_k$, but instead observe $\mathbf{y}_k = h(\mathbf{x}_k)$ for some unknown function $h$.

We have available a set of training data consisting of multiple sequences of length $L + T$. The inference task is to forecast the values $\mathbf{y}_{L+1}, \ldots, \mathbf{y}_{L+T}$ given observations of the first $L$ values of a sequence $\mathbf{y}_1, \ldots, \mathbf{y}_L$.

### 3.2. Strategy: Directly observable systems

We first consider the setting where we directly observe the state, i.e., $h$ is the identity, and $\mathbf{y}_k = h(\mathbf{x}_k) = \mathbf{x}_k$. Assume that the dynamical system can be captured by $\mathbf{x}_{k+1} = F(\mathbf{x}_k)$. Then an appropriate forecasting approach is to learn a finite dimensional approximation to the Koopman operator associated with $F$, and then propagate it forward in time to construct the forecast. In this section, for notational simplicity, we describe a setting where learning is based on a single observation $\mathbf{y}_0, \ldots, \mathbf{y}_L$ (which, for now, $= \mathbf{x}_0, \ldots, \mathbf{x}_L$). The extension to learning using multiple observed series is straightforward.

Let us introduce measurement functions $g_1, g_2, \ldots g_{n_g} \in \mathcal{H}$, where $\mathcal{H}$ is a Hilbert space containing real-valued functions defined on $\mathcal{X}$. Denoting $\mathcal{L}(\mathcal{H})$ as the space of bounded linear operators $T : \mathcal{H} \to \mathcal{H}$, Khosravi (2023) formulates the learning of the Koopman operator as a minimization

task with a Tikhonov-regularized empirical loss:

$$\min_{K \in \mathcal{L}(\mathcal{H})} \sum_{k=1}^{L} \sum_{l=1}^{n_g} \left( \mathbf{x}_{kl} - (Kg_l)(\mathbf{x}_{k-1}) \right)^2 + \lambda ||K||^2 . \quad (6)$$

Although this optimization is over an infinite-dimensional space of linear operators, Khosravi (2023) demonstrates that if the measurement functions satisfy certain conditions, then there is a unique solution $\hat{K}$, which can be derived by solving a finite-dimensional optimization problem.

A special case occurs when $\hat{K}$ is invariant in the subspace spanned by the measurement functions, $\mathcal{G} = \text{span}\{g_1, \ldots, g_{n_g}\}$, i.e. $\hat{K} \in \mathcal{L}(\mathcal{G})$. In this setting, we can follow the approach of the Extended Dynamic Mode Decomposition (EDMD) method (Li et al., 2017), which approximates the Koopman operator by a finite-dimensional linear map $U : \mathcal{G} \to \mathcal{G}$. We can represent the action of U on $g_l$ using a matrix $\mathbf{M} \in \mathbb{R}^{n_g \times n_g}$. Because the dimension of $\mathcal{G}$ is finite, we can identify an $\mathbf{M}$ such that $Ug_l = \sum_{j=1}^{n_g} [M]_{j,l} g_j$. The matrix $\mathbf{M}$ can then be estimated via the following minimization:

$$\min_{\mathbf{M}} ||\mathbf{P}_G \mathbf{M} - \mathbf{Y}||_F^2 . \quad (7)$$

where $\mathbf{P}_G = [g_l(\mathbf{x}_{k-1})]_{k=1,l=1}^{L,n_g}$ and $\mathbf{Y} = [g_l(\mathbf{x}_k)]_{k=1,l=1}^{L,n_g}$. When applying EDMD, we assume that $\mathbf{P}_G$ is full rank so that a unique minimizer can be identified via the Moore-Penrose pseudoinverse.

This learning task can be made more flexible by allowing for learning of the measurement functions $g_l$. Li et al. (2017) propose a method that incorporates neural network based learning of the measurement functions and an $L_1$ regularizer to promote sparsity.

### 3.3. Unobserved states

In many settings, we do not observe the state of the system $\mathbf{x}_k$ directly. Instead we observe $\mathbf{y}_k = h(\mathbf{x}_k)$ for some unknown observation function $h(\cdot)$. In this setting, $\mathbf{y}_k$ may not provide sufficient information in isolation to recover the state $\mathbf{x}_k$. We can instead construct a state representation by considering the past $L$ measurements, i.e., we define $\widetilde{\mathbf{x}}_k = [\mathbf{y}_{k-L+1}, \ldots, \mathbf{y}_k]^\top$. With this representation, we can model the dynamics as $\widetilde{\mathbf{x}}_k = \tilde{F}(\widetilde{\mathbf{x}}_{k-1})$ and target learning of the Koopman operator $\hat{K}$ associated with $\tilde{F}$. Note that this permits us to perform prediction, because we are interested in predicting $\mathbf{y}_{k+1}$, which can be recovered from $\widetilde{\mathbf{x}}_{k+1}$.

Under the same assumptions of invariance of the Koopman operator with respect to $\mathcal{G}$, we can adopt the same approach as outlined above, learning a matrix $\mathbf{M}$. Given the structure of the constructed $\widetilde{\mathbf{x}}_k$, we are motivated to impose further structure on the Koopman operator matrix, with the goal of introducing an inductive bias that can facilitate learning and

make it more robust. In particular, we enforce a structure on $\mathbf{M}$ that allows us to write:

$$g(\widetilde{\mathbf{x}}_k) = \mathbf{M}g(\widetilde{\mathbf{x}}_{k-1}) = g(\mathbf{y}_{k-1}) + \sum_{s=1}^{L} \mathbf{W}^s g(\mathbf{y}_{k-s}) . \quad (8)$$

With this structure, we see that $\mathbf{M}$ is a blockwise diagonal matrix, where each block is a power of a learnable matrix $\mathbf{W}$. Moreover, by comparing Eq. 8 with Eq. 5, we see that this structure, which represents the dynamic state using a collection of time-delayed observations (Arbabi & Mezic, 2017), can be implemented as a linear RNN.

### 3.4. SKOLR

Building on our analysis of Koopman operator approximation and the connection to the linear RNN, we present SKOLR, which integrates a learnable spectral decomposition of the input signal with a multilayer perceptron (MLP) for the measurement functions.

Inspired by multiresolution DMD (Kutz et al., 2016), instead of learning a single linear RNN acting on a high dimensional space, we propose to split the space into multiple subspaces, resulting in learning a structured Koopman operator via a highly parallel linear RNN stack. This structure also improves the parameter efficiency, as shown in Fig. 1

**Encoder**   Let the input sequence be $\mathbf{Y} = [\mathbf{y}_1, \mathbf{y}_2, \ldots, \mathbf{y}_L]$, where $\mathbf{y}_k \in \mathbb{R}^P$, with $P$ being the dimension of the observation. The encoder performs learnable frequency decomposition via reconstruction of the soft-gated frequency spectrum via Fast Fourier Transform (FFT) and Inverse FFT (IFFT):

$$\begin{aligned} \mathbf{S} &= \text{FFT}(\mathbf{Y}), \\ \mathbf{S}_n &= \mathbf{S} \cdot \text{Sigmoid}(\mathbf{w}_n), \quad (9) \\ \mathbf{Y}_n &= \text{IFFT}(\mathbf{S}_n). \end{aligned}$$

The reconstructed signals $\{\mathbf{Y}_n\}_{n=1}^N$ form $N$ parallel branches, with each branch representing a frequency-based subspace, while $\{\mathbf{w}_n\}_{n=1}^N$ contain learnable parameters for frequency selection.

For each branch $n$ we parameterize the measurement functions using non-linear feed-forward network:

$$\mathbf{z}_{n,k} = \text{FFN}_{\text{enc},n}(\mathbf{y}_k), \text{ for } k = 1, \ldots, L \quad (10)$$

where FFN $: \mathbb{R}^P \to \mathbb{R}^D$. We utilize multiple layer perceptrons (MLPs) for simplicity, generally with only one layer or two. Other FFNs like wiGLU (Shazeer, 2020) are also applicable. After this encoding is complete, we have constructed the $\mathbf{z}_k = g(\mathbf{y}_k)$ (and hence the $g(\widetilde{\mathbf{x}}_k)$) that appear in Eq. 8 for $k = 1, \ldots, L$. By structuring the architecture into multiple branches and incorporating both frequency-domain filtering and time-domain encoding, we enhance flexibility in learning suitable measurement functions for diverse time-series patterns.

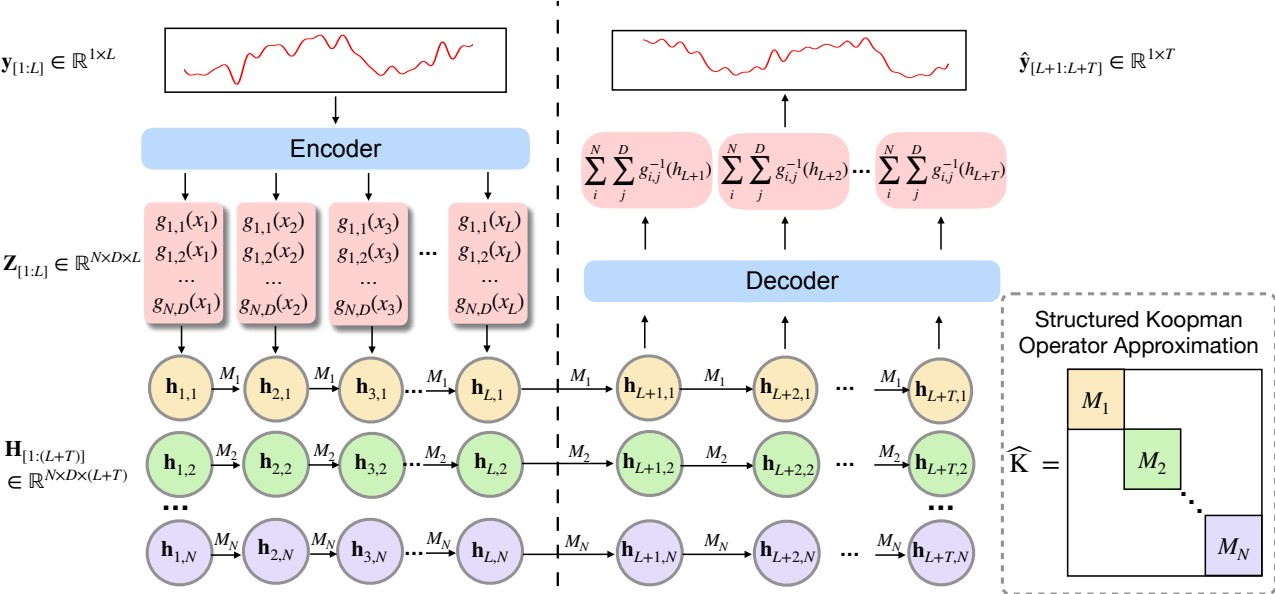

Figure 1: Architecture of SKOLR (**S**tructured **K**oopman **O**perator **L**inear **R**NN) The input time series goes through an encoder with learnable frequency decomposition and a MLP that models the measurement functions. With the branch decomposition, the highly parallel linear RNN chains jointly attend to different dynamical patterns from different representation subspaces. Finally, a decoder reconstructs predictions by parameterizing the inverse measurement functions. This structured approach maintains computational efficiency while naturally aligning with Koopman principles.

**RNN Stack** Given the collection for each branch $n$ and time step $k$: $\mathbf{Z}_n = [\mathbf{z}_{1,n}, \dots, \mathbf{z}_{L,n}] \in \mathbb{R}^{D \times L}$, we take it as input to a linear RNN and introduce learnable branch-specific weight matrices $\mathbf{W}_n$ for each branch.

$$\mathbf{h}_{k+1,n} = \mathbf{W}_n \mathbf{h}_{k,n} + \mathbf{z}_{k,n} \qquad (11)$$

Each branch weight matrix $\mathbf{W}_n$ defines a matrix $\mathbf{M}_n$, as discussed above, which specifies a finite-dimensional approximation to a Koopman operator for $\widetilde{\mathbf{x}}$ for the learned measurement functions on that branch.

Together, the branch matrices $\mathbf{M}_n$ form a structured finite-dimensional Koopman operator approximation $\widehat{\mathrm{K}}$ with block diagonal structure:

$$\widehat{\mathrm{K}} = \begin{bmatrix} \mathbf{M}_1 & 0 & \cdots & 0 \\ 0 & \mathbf{M}_2 & \cdots & 0 \\ \vdots & \vdots & \ddots & \vdots \\ 0 & 0 & \cdots & \mathbf{M}_N \end{bmatrix} \qquad (12)$$

By imposing this structure and using a stack of linear RNNs, we can learn local approximations to the evolution dynamics of different observables. For each branch $n$:

$$\mathbf{H}_n = [\mathbf{z}_{1,n}, \mathbf{z}_{2,n} + \mathbf{M}_n \mathbf{z}_{1,n}, \dots, \mathbf{z}_{L,n} + \sum_{s=0}^{L-1} \mathbf{M}_n^s \mathbf{z}_{s,n}] \qquad (13)$$

For prediction of length $T$, we recursively apply the operator to predict the Koopman space for future steps per branch:

$$\mathbf{H}_{[L+1:L+T],n} = [\mathbf{M}_n \mathbf{h}_{L,n}, \dots, \mathbf{M}_n^T \mathbf{h}_{L,n}] \qquad (14)$$

**Decoder** For reconstruction, we use mirrored feed-forward networks to parameterize the inverse measurement functions $g^{-1}$. The decoder processes the hidden states as:

$$\hat{\mathbf{y}}_{k,n} = \mathrm{FFN}_{\mathrm{dec},n}(\mathbf{h}_{k,n}) \qquad (15)$$

where $\mathrm{FFN}_{\mathrm{dec}} : \mathbb{R}^D \to \mathbb{R}^P$.

The decoder combines predictions from all branches to generate the final prediction $\hat{\mathbf{y}}_{[L+1,L+T]}$. The model is trained end-to-end using the loss function:

$$\mathcal{L} = \|\hat{\mathbf{y}}_{[L+1:L+T]} - \mathbf{y}_{[L+1:L+T]}\|_2^2. \qquad (16)$$

The structured approach, induced by both the linear RNN and the branch decomposition, enables efficient parallel processing and reduces the parameter count. Since all architectural components are very simple (basic sigmoid frequency gating, one- or two-layer MLPs for encoding/decoding, linear RNN), the architecture is very fast to train and has low memory cost, as we illustrate in the experiments section.

Table 1: Prediction results on benchmark datasets, $L = 2T$ and $T \in \{48, 96, 144, 192\}$ (ILI: $T \in \{24, 36, 48, 60\}$). Best results and second best results are highlighted in red and blue respectively.

| Models | T | SKOLR | | Koopa | | iTransformer | | PatchTST | | TimesNet | | Dlinear | | MICN | | KNF | | Autoformer | |
|---|---|---|---|---|---|---|---|---|---|---|---|---|---|---|---|---|---|---|---|
| | | MSE | MAE | MSE | MAE | MSE | MAE | MSE | MAE | MSE | MAE | MSE | MAE | MSE | MAE | MSE | MAE | MSE | MAE |
| ECL | 48 | 0.137 | 0.229 | 0.130 | 0.234 | 0.133 | 0.225 | 0.147 | 0.246 | 0.149 | 0.254 | 0.158 | 0.241 | 0.156 | 0.271 | 0.175 | 0.265 | 0.164 | 0.272 |
| | 96 | 0.132 | 0.225 | 0.136 | 0.236 | 0.134 | 0.230 | 0.143 | 0.241 | 0.170 | 0.275 | 0.153 | 0.245 | 0.165 | 0.277 | 0.198 | 0.284 | 0.182 | 0.289 |
| | 144 | 0.143 | 0.236 | 0.149 | 0.247 | 0.145 | 0.240 | 0.145 | 0.241 | 0.183 | 0.287 | 0.152 | 0.245 | 0.163 | 0.274 | 0.204 | 0.297 | 0.210 | 0.315 |
| | 192 | 0.149 | 0.244 | 0.156 | 0.254 | 0.154 | 0.249 | 0.147 | 0.240 | 0.189 | 0.291 | 0.153 | 0.246 | 0.171 | 0.284 | 0.245 | 0.321 | 0.221 | 0.324 |
| Traffic | 48 | 0.400 | 0.258 | 0.415 | 0.274 | 0.369 | 0.256 | 0.426 | 0.286 | 0.567 | 0.306 | 0.488 | 0.352 | 0.496 | 0.301 | 0.621 | 0.382 | 0.640 | 0.361 |
| | 96 | 0.368 | 0.248 | 0.401 | 0.275 | 0.388 | 0.270 | 0.413 | 0.283 | 0.611 | 0.337 | 0.485 | 0.336 | 0.511 | 0.312 | 0.645 | 0.376 | 0.668 | 0.367 |
| | 144 | 0.375 | 0.255 | 0.397 | 0.276 | 0.375 | 0.267 | 0.405 | 0.278 | 0.603 | 0.322 | 0.452 | 0.317 | 0.498 | 0.309 | 0.683 | 0.402 | 0.681 | 0.379 |
| | 192 | 0.377 | 0.256 | 0.403 | 0.284 | 0.373 | 0.267 | 0.404 | 0.277 | 0.604 | 0.321 | 0.438 | 0.309 | 0.494 | 0.312 | 0.699 | 0.405 | 0.692 | 0.385 |
| Weather | 48 | 0.131 | 0.170 | 0.126 | 0.168 | 0.136 | 0.174 | 0.140 | 0.179 | 0.138 | 0.191 | 0.156 | 0.198 | 0.157 | 0.217 | 0.201 | 0.288 | 0.185 | 0.240 |
| | 96 | 0.154 | 0.202 | 0.154 | 0.205 | 0.169 | 0.216 | 0.160 | 0.206 | 0.180 | 0.231 | 0.186 | 0.229 | 0.187 | 0.250 | 0.295 | 0.308 | 0.230 | 0.279 |
| | 144 | 0.172 | 0.220 | 0.172 | 0.225 | 0.192 | 0.242 | 0.174 | 0.221 | 0.190 | 0.244 | 0.199 | 0.244 | 0.197 | 0.257 | 0.394 | 0.401 | 0.268 | 0.308 |
| | 192 | 0.193 | 0.241 | 0.193 | 0.241 | 0.204 | 0.251 | 0.195 | 0.243 | 0.212 | 0.265 | 0.217 | 0.261 | 0.214 | 0.270 | 0.462 | 0.437 | 0.325 | 0.347 |
| ETTm1 | 48 | 0.280 | 0.330 | 0.283 | 0.333 | 0.313 | 0.356 | 0.286 | 0.336 | 0.308 | 0.354 | 0.322 | 0.355 | 0.294 | 0.353 | 1.026 | 0.792 | 0.592 | 0.419 |
| | 96 | 0.289 | 0.340 | 0.294 | 0.345 | 0.302 | 0.353 | 0.299 | 0.346 | 0.329 | 0.370 | 0.309 | 0.346 | 0.306 | 0.364 | 0.957 | 0.782 | 0.493 | 0.469 |
| | 144 | 0.319 | 0.361 | 0.322 | 0.366 | 0.331 | 0.374 | 0.325 | 0.363 | 0.358 | 0.387 | 0.327 | 0.359 | 0.342 | 0.390 | 0.921 | 0.760 | 0.735 | 0.569 |
| | 192 | 0.328 | 0.373 | 0.337 | 0.378 | 0.343 | 0.381 | 0.343 | 0.375 | 0.462 | 0.441 | 0.337 | 0.365 | 0.386 | 0.415 | 0.896 | 0.731 | 0.592 | 0.506 |
| ETTm2 | 48 | 0.134 | 0.228 | 0.134 | 0.226 | 0.139 | 0.234 | 0.135 | 0.231 | 0.142 | 0.234 | 0.144 | 0.240 | 0.131 | 0.238 | 0.621 | 0.623 | 0.191 | 0.280 |
| | 96 | 0.171 | 0.255 | 0.171 | 0.254 | 0.177 | 0.268 | 0.171 | 0.255 | 0.187 | 0.269 | 0.172 | 0.256 | 0.197 | 0.295 | 1.535 | 1.012 | 0.241 | 0.311 |
| | 144 | 0.209 | 0.283 | 0.206 | 0.280 | 0.216 | 0.296 | 0.205 | 0.282 | 0.216 | 0.291 | 0.200 | 0.276 | 0.210 | 0.297 | 1.337 | 0.876 | 0.300 | 0.352 |
| | 192 | 0.241 | 0.304 | 0.226 | 0.298 | 0.237 | 0.310 | 0.221 | 0.294 | 0.243 | 0.313 | 0.219 | 0.290 | 0.248 | 0.328 | 1.355 | 0.908 | 0.324 | 0.370 |
| ETTh1 | 48 | 0.333 | 0.373 | 0.336 | 0.377 | 0.342 | 0.380 | 0.337 | 0.375 | 0.365 | 0.399 | 0.343 | 0.371 | 0.375 | 0.406 | 0.876 | 0.709 | 0.442 | 0.438 |
| | 96 | 0.371 | 0.398 | 0.371 | 0.405 | 0.393 | 0.412 | 0.372 | 0.393 | 0.411 | 0.430 | 0.379 | 0.393 | 0.406 | 0.429 | 0.975 | 0.744 | 0.634 | 0.523 |
| | 144 | 0.405 | 0.417 | 0.405 | 0.418 | 0.425 | 0.430 | 0.394 | 0.412 | 0.442 | 0.447 | 0.393 | 0.403 | 0.437 | 0.448 | 0.801 | 0.662 | 0.522 | 0.491 |
| | 192 | 0.422 | 0.432 | 0.416 | 0.429 | 0.456 | 0.454 | 0.416 | 0.412 | 0.469 | 0.470 | 0.407 | 0.416 | 0.518 | 0.496 | 0.941 | 0.744 | 0.525 | 0.501 |
| ETTh2 | 48 | 0.238 | 0.306 | 0.226 | 0.300 | 0.243 | 0.313 | 0.223 | 0.297 | 0.241 | 0.319 | 0.226 | 0.305 | 0.260 | 0.336 | 0.385 | 0.376 | 0.355 | 0.380 |
| | 96 | 0.299 | 0.352 | 0.297 | 0.349 | 0.306 | 0.358 | 0.300 | 0.353 | 0.325 | 0.376 | 0.294 | 0.351 | 0.343 | 0.393 | 0.433 | 0.446 | 0.427 | 0.432 |
| | 144 | 0.335 | 0.377 | 0.333 | 0.381 | 0.347 | 0.385 | 0.346 | 0.390 | 0.374 | 0.408 | 0.354 | 0.397 | 0.374 | 0.411 | 0.441 | 0.456 | 0.457 | 0.461 |
| | 192 | 0.365 | 0.397 | 0.356 | 0.393 | 0.375 | 0.403 | 0.383 | 0.406 | 0.394 | 0.434 | 0.385 | 0.418 | 0.455 | 0.464 | 0.528 | 0.503 | 0.503 | 0.491 |
| ILI | 24 | 1.556 | 0.760 | 1.621 | 0.800 | 1.763 | 0.843 | 2.063 | 0.881 | 2.464 | 1.039 | 2.624 | 1.118 | 4.380 | 1.558 | 3.722 | 1.432 | 2.831 | 1.085 |
| | 36 | 1.462 | 0.728 | 1.803 | 0.855 | 2.067 | 0.919 | 2.178 | 0.943 | 2.388 | 1.007 | 2.693 | 1.156 | 3.314 | 1.313 | 3.941 | 1.448 | 2.801 | 1.088 |
| | 48 | 1.537 | 0.798 | 1.768 | 0.903 | 1.667 | 0.879 | 1.916 | 0.896 | 2.370 | 1.040 | 2.852 | 1.229 | 2.457 | 1.085 | 3.287 | 1.377 | 2.322 | 1.006 |
| | 60 | 2.187 | 0.995 | 1.743 | 0.891 | 2.011 | 1.000 | 1.981 | 0.917 | 2.193 | 1.003 | 2.554 | 1.144 | 2.379 | 1.040 | 2.974 | 1.301 | 2.470 | 1.061 |
| Rank 1st | # | 17 | 15 | 10 | 7 | 3 | 2 | 3 | 3 | 0 | 0 | 5 | 7 | 1 | 0 | 0 | 0 | 0 | 0 |

# 4. Experiments

## 4.1. Benchmarking SKOLR

### 4.1.1. DATASETS

We evaluate SKOLR on widely-used public benchmark datasets. For long-term forecasting, we use Weather, Traffic, Electricity, ILI and four ETT datasets (ETTh1, ETTh2, ETTm1, ETTm2). We assess short-term performance on M4 dataset (Makridakis et al., 2020), which includes six subsets of periodically recorded univariate marketing data. For more information about the datasets see Appendix A.1.

### 4.1.2. BASELINES AND EXPERIMENTAL SETTINGS

We compare against state-of-the-art deep forecasting models. The comparison includes transformer-based models: Autoformer (Wu et al., 2021), PatchTST (Nie et al., 2023), iTransformer (Liu et al., 2024), TCN-based models: TimesNet (Wu et al., 2023), MICN (Wang et al., 2023a), linear model: DLinear (Zeng et al., 2023), and Koopman-based models: KNF (Wang et al., 2023b), Koopa (Liu et al., 2023).

We select these representative baselines for their established performance and public implementations.

Following Koopa (Liu et al., 2023), we set the lookback window length $L = 2T$ for prediction horizon $T \in \{48, 96, 144, 192\}$ for all datasets, except ILI, for which we use $T \in \{24, 36, 48, 60\}$. This setting leverages more historical data for longer forecasting horizons. We report baseline results from Liu et al. (2023) except for iTransformer; we reproduce iTransformer results with $L = 2T$ using the officially released code. Performance is measured using Mean Squared Error (MSE) and Mean Absolute Error (MAE). Appendix A.2 provides implementation details.

### 4.1.3. RESULTS AND ANALYSIS

Table 1 reports the experimental results for eight benchmarks. The performance is measured by MSE and MAE; the best and second-best results for each case (dataset, horizon, and metric) are highlighted in bold and underlined, respectively. The results are the average of 3 trials.

We rank the algorithms in Table 1 based on their MSE

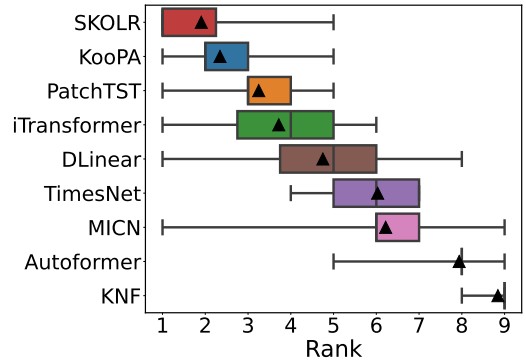

Figure 2: Boxplot for ranks of the algorithms (based on their MSE) across seven datasets and four prediction horizons. The medians and means of the ranks are shown by the vertical lines and the black triangles respectively; whiskers extend to the minimum and maximum ranks.

Table 2: Model evaluation results (MSE/MAE) on non-linear dynamical systems (NLDS)

| Dataset | SKOLR | | KooPA | |
|---|---|---|---|---|
| | MSE | MAE | MSE | MAE |
| Pendulum | **0.0001** | **0.0083** | 0.0039 | 0.0470 |
| Duffing | **0.0047** | **0.0518** | 0.0365 | 0.1479 |
| Lotka-Volterra | **0.0018** | **0.0354** | 0.0178 | 0.1050 |
| Lorenz '63 | **0.9740** | **0.7941** | 1.0937 | 0.8325 |

and order them based on their average rank across eight datasets and four prediction horizons. Figure 2 shows the relative ranks. We observe that SKOLR achieves SOTA performance, with the best average rank across all settings.

The model shows strength in capturing complex patterns in the Weather dataset, matching Koopa's performance while surpassing other transformers, indicating effective handling of meteorological dynamics. For the ILI dataset, which features highly nonlinear epidemic patterns, SKOLR outperforms the baseline methods, with significant error reduction for the shorter horizons.

While SKOLR demonstrates strong performance in long-term forecasting, we also evaluate its effectiveness on short-term predictions with M4 dataset. Results in Appendix B.1 show consistent improvements over both transformer-based forecasting methods and Koopman-based alternatives across different time scales.

## 4.2. State Prediction for Non-Linear Systems

Koopman operator-based approaches have gained attention for their ability to perform system identification in a fully data-driven manner. To evaluate SKOLR's performance in this context, we conducted a series of experiments on non-linear dynamical systems (NLDS) (details in Appendix E).

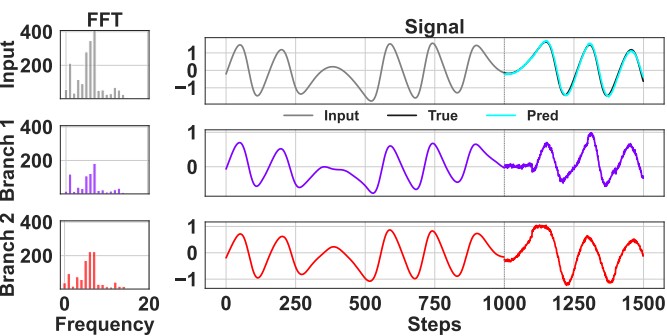

Figure 3: Analysis of SKOLR's branch-wise behavior: (a) frequency decomposition and (b) prediction performance. We observe that different branches focus on different frequency components.

Table 2 demonstrates SKOLR's effectiveness across different dynamical systems. For periodic systems like Pendulum, SKOLR achieves substantial improvements, indicating superior capture of oscillatory patterns. In chaotic systems like Lorenz '63, SKOLR shows better stability with 10.9% reduction on MSE, suggesting robust handling of sensitive dependence on initial conditions. The model demonstrates particularly strong performance on mixed dynamics: Lotka-Volterra and Duffing oscillator. These results validate that SKOLR's structured operator design effectively captures both periodic motions and complex nonlinear dynamics.

Fig. 3 demonstrates SKOLR's multi-scale decomposition strategy. The FFT analysis reveals how different branches place more emphasis on some frequency bands. This natural frequency partitioning emerges from our structured Koopman design, enabling each branch to focus on specific temporal scales. The prediction visualization illustrates the complementary nature of these branches, where their combined forecasts reconstruct complex dynamics through principled superposition of simpler, frequency-specific predictions. More analysis can be found in Appendix D.3.

## 4.3. Analysis and Ablation Study

### 4.3.1. ANALYSIS: STRUCTURED KOOPMAN OPERATOR

We analyze the impact of branch configurations through two controlled experiments: (1) Fixed parameter count scenario, where total parameters remain constant ($\sim$1.6M) across configurations while varying the learnable frequency decomposition $\mathbf{w}$, with $N$ branches and lookback window $L$; (2) Fixed dimension scenario: We maintain constant Koopman operator approximation dimension $\dim(\widehat{\mathrm{K}}) = 512$ while varying branch number $N$. As $N$ increases, each branch's dimension $D$ decreases proportionally ($D = 512/N$), leading to reduced parameter count.

We conduct experiments on the ETTm1 dataset. As we can see in Table 3 and Fig. 4, maintaining similar parameter

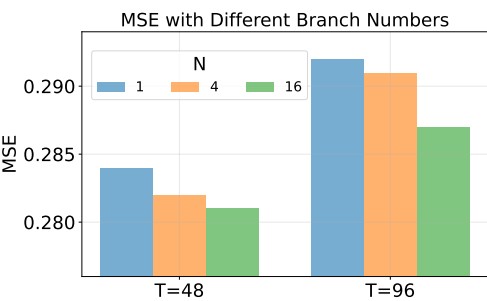

Figure 4: MSE comparison on ETTm1 dataset across different branch configurations and prediction horizons. Bars show MSE values for each configuration. All configurations maintain similar parameter counts (∼1.6M). Increasing branch number improves performance.

Table 3: Performance comparison with similar parameter counts on ETTm1

| Config. | MSE for Different $T$ | | | |
|---|---|---|---|---|
| (D, N) | 48 | 96 | 144 | 192 |
| (512, 1) | 0.284 | 0.292 | 0.326 | 0.330 |
| (256, 4) | 0.282 | 0.291 | **0.317** | 0.341 |
| (128, 16) | **0.281** | **0.287** | 0.323 | **0.329** |

counts (∼1.6M) and increasing branch numbers from $N = 1$ to $N = 16$ improves performance for most horizons.

More significantly, when keeping $\dim(\widehat{K}) = 512$ (Table 4), models with more branches maintain strong performance despite substantial parameter reduction. Notably, the configuration with $D = 32, N = 16$ achieves comparable performance to the 1-branch model while using only 0.25M parameters (85% reduction). This demonstrates that structured decomposition through multiple branches enables significantly more efficient parameter utilization while maintaining or improving forecasting accuracy.

### 4.3.2. ABLATION: IMPACT OF FREQUENCY DECOMPOSITION

The improved performance with multiple branches motivates further analysis of our learnable frequency decomposition strategy. In Equation 9, the learnable matrix $\mathbf{w}$ enables adaptive frequency allocation across branches, in contrast to uniform decomposition ($\mathbf{w} = \mathbf{1}$). Table 5 demonstrates that this learnable approach consistently outperforms uniform allocation across prediction horizons.

This adaptive capability is particularly beneficial for datasets with complex temporal patterns (Weather, ECL), where different frequency bands may carry varying importance at different time scales. The learned masks show distinct patterns across datasets, suggesting that the model successfully adapts its frequency decomposition strategy based on the un-

Table 4: Performance comparison with $\dim(\widehat{K}) = 512$ and varying branch numbers $N$ on ETTm1. Parameter counts shown for horizon $T = 192$.

| Config. | Params | MSE for Different $T$ | | | |
|---|---|---|---|---|---|
| (D, N) | (M) | 48 | 96 | 144 | 192 |
| (512, 1) | 1.71 | 0.284 | **0.292** | 0.326 | 0.330 |
| (256, 2) | 0.92 | **0.280** | 0.294 | 0.318 | 0.334 |
| (128, 4) | 0.53 | **0.280** | 0.293 | 0.318 | 0.335 |
| (64, 8) | 0.34 | 0.283 | 0.297 | 0.316 | 0.329 |
| (32, 16) | 0.25 | 0.282 | **0.292** | **0.313** | **0.328** |

Table 5: Ablation Study on frequency decomposition

| Dataset | T | SKOLR (learn) | | SKOLR($\mathbf{w} = 1$) | |
|---|---|---|---|---|---|
| | | MSE | MAE | MSE | MAE |
| ECL | 48 | **0.137** | **0.229** | 0.150 | 0.239 |
| | 96 | **0.132** | **0.225** | 0.134 | 0.227 |
| | 144 | **0.143** | **0.236** | 0.144 | 0.237 |
| | 192 | **0.149** | 0.244 | 0.150 | 0.244 |
| Weather | 48 | **0.131** | **0.170** | 0.134 | 0.173 |
| | 96 | **0.154** | **0.202** | 0.158 | 0.203 |
| | 144 | **0.172** | **0.220** | 0.175 | 0.221 |
| | 192 | **0.193** | **0.241** | 0.194 | 0.242 |
| ETTh1 | 48 | 0.333 | 0.373 | **0.329** | **0.371** |
| | 96 | **0.371** | **0.398** | 0.375 | 0.400 |
| | 144 | **0.405** | **0.417** | 0.407 | 0.419 |
| | 192 | **0.422** | **0.432** | 0.429 | 0.433 |

derlying data characteristics. Notably, on the ETTh1 dataset, learnable decomposition occasionally underperforms uniform masking, particularly at shorter horizons ($T = 48$). This suggests potential overfitting on smaller datasets.

### 4.4. Model Efficiency

To demonstrate the computational efficiency of SKOLR, we analyze the model complexity in terms of parameter count, GPU Memory and Running Time. We compare these values with several baseline models on the ETTm1 and weather dataset with sequence length 96 and prediction length 48.

Fig. 5 demonstrates SKOLR's computational advantages. On ETTm1, SKOLR achieves the best MSE while using only 3.31 MiB GPU memory. The training speed is also notably faster than other methods. On Weather dataset, SKOLR maintains competitive accuracy, while using significantly less memory and training 4x faster compared to the best. This exceptional efficiency-performance trade-off stems from our structured linear operations in Koopman space, avoiding the quadratic complexity of self-attention while maintaining modeling capacity through parallel branch architecture. The computational efficiency for all datasets can be found in Appendix C.

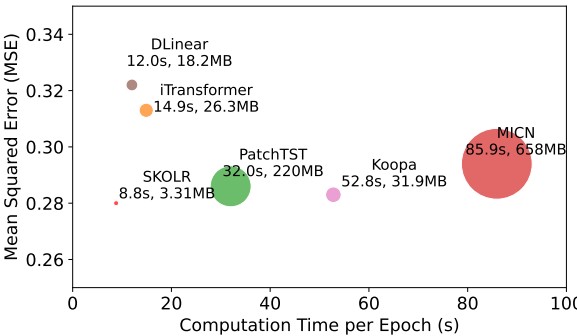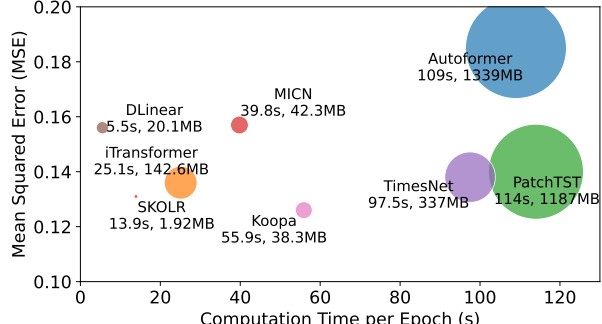

Figure 5: Model comparison on error and training epoch time on P100 GPU. Memory consumption is proportional to circle radius. On ETTm1 (left) we observe that SKOLR is both the fastest and most accurate method while requiring the smallest memory footprint. On Weather (right) SKOLR is the second best method with much lower training memory and time consumption than the best. For an equivalent budget one could train an ensemble of our approach and obtain better results.

## 5. Related Work

### 5.1. Koopman Operator-based Time-series Forecasting

Koopman theory (Koopman, 1931) has been applied for modeling and analyzing complex dynamical systems for decades (Mezić, 2005; Brunton et al., 2022). The major advantage of the Koopman operator is that it can represent the dynamical system in the form of a linear operator acting on measurement functions (observables). However, learning the operator is challenging because it has infinite dimension. Researchers strive to develop effective strategies for performing finite dimensional approximations; key to this is the selection of good measurement functions. To address this, recent work has explored neural networks for learning the mapping and the approximate operator simultaneously (Li et al., 2017; Lusch et al., 2018; Takeishi et al., 2017; Yeung et al., 2019).

Three recent works address time-series forecasting using Koopman operators. K-Forecast (Lange et al., 2021) uses Koopman theory to handle the nonlinearity in temporal signals and proposes a data-dependent basis for long-term time-series forecasting. By leveraging predefined measurement functions, KNF (Wang et al., 2023b) learns the Koopman operator and attention map to cope with time-series forecasting with changing temporal distributions. Koopa (Liu et al., 2023) introduces modular Koopman predictors that separately address time-variant and time-invariant components via a hierarchical architecture, using learnable operators for the latter and eDMD (Williams et al., 2015) for the former. These prior works rely on hierarchical architectures or complex spectral decompositions to approximate Koopman operators. Our work takes a different approach, drawing a connection with linear RNNs, paving the way to a very efficient and simple forecasting architecture. Our results demonstrate that this strategy leads to improved accuracy with reduced computational overhead and memory.

Although Orvieto et al. (2023) provided insights into the potential connections between the Koopman operator and a wide MLP + linear RNN for representing dynamical systems, this was not the primary focus of their work, and they did not provide equations demonstrating the connection or conduct empirical verification. In this work, building on similar insights, we establish an explicit connection by deriving equations that demonstrate a direct analogy between a structured approximation of a Koopman operator and an architecture consisting of an MLP encoder combined with a linear RNN.

### 5.2. Deep Learning for Time-Series Forecasting

Time-series forecasting has evolved from statistical models (Makridakis & Hibon, 1997; Hyndman et al., 2008) to deep learning approaches. Previous methods used RNNs (Salinas et al., 2020; Smyl, 2020; Mienye et al., 2024) and CNNs (Bai et al., 2018; Luo et al., 2024) for their ability to capture temporal dependencies. MLP-based architectures (Oreshkin et al., 2020; Challu et al., 2023; Vijay et al., 2023; Wang et al., 2024a) also demonstrated promising performance for forecasting. Recently, transformer architectures (Nie et al., 2023; Zhang et al., 2024; Hounie et al., 2024; Ilbert et al., 2024) introduced powerful attention mechanisms, with innovations in basis functions (Ni et al., 2024) and channel-wise processing (Liu et al., 2024). To address their quadratic complexity, sparse attention variants (Lin et al., 2024) were proposed, but these often struggle with capturing long-range dependencies due to information loss from pruned attention scores. Foundation models (Das et al., 2024; Darlow et al., 2024) and unified approaches (Woo et al., 2024) have recently emerged. These attempt to mitigate the limitations through pre-training and multi-task learning, but this comes at the cost of dramatically increased architectural complexity and computational overhead. To address the complexity challenges in time-

series forecasting, recent state space models (Gu & Dao, 2023) achieve linear complexity, while physics-informed approaches (Verma et al., 2024) enhance interpretability. However, these methods often require complex architectures or domain expertise. Our approach offers a balanced solution with a principled foundation based on Koopman theory, achieving excellent prediction performance with very low computation and memory requirements.

## 6. Conclusion

This work establishes a connection between Koopman operator approximation and linear RNNs, showing that time-delayed state representations yield an equivalence between structured Koopman operators and linear RNN updates. Based on this, we introduce SKOLR, which integrates learnable spectral decomposition with a parallelized linear RNN stack giving rise to the structured Koopman operator. By aligning deep learning with Koopman theory, this approach provides a principled and computationally efficient solution for nonlinear time-series modeling. Empirical evaluations on forecasting benchmarks and dynamical systems demonstrate that SKOLR achieves strong predictive performance while maintaining the efficiency of linear RNNs. Future work includes extending this framework to broader dynamical systems and exploring alternative spectral representations for enhanced expressivity.

## Acknowledgement

This research was funded by the Natural Sciences and Engineering Research Council of Canada (NSERC), [reference number 260250]. Cette recherche a été financée par le Conseil de recherches en sciences naturelles et en génie du Canada (CRSNG), [numéro de référence 260250]. Ce projet de recherche #324302 est rendu possible grâce au financement du Fonds de recherche du Québec.

## Impact Statement

This paper presents work whose goal is to advance the field of Machine Learning. There are many potential societal consequences of our work, none which we feel must be specifically highlighted here.

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

# A. Experimental Details

## A.1. Dataset

We evaluate the performance of our proposed SKOLR on eight widely-used public benchmark datasets, including Weather, Traffic, Electricity, ILI and four ETT datasets (ETTh1, ETTh2, ETTm1, ETTm2). *Weather* is a collection of 2020 weather data from 21 meteorological indicators, including air temperature and humidity, provided by the Max-Planck Institute for Biogeochemistry. [2] *Traffic* is a dataset provided by Caltrans Performance Measurement System (PeMS), collecting hourly data of the road occupancy rates measured by different sensors on San Francisco Bay area freeways from California Department of Transportation. [3] *Electricity* contains hourly time series of the electricity consumption of 321 customers from 2012 to 2014 (Trindade, 2015; Wu et al., 2021). [4] *ILI* dataset[5] contains the weekly time series of ratio of patients seen with ILI and the total number of the patients in the United States between 2002 and 2021. *ETT* datasets are a series of measurements, including load and oil temperature, from electricity transformers between 2016 and 2018, provided by Zhou et al. (2021). Following the standard pipelines, the dataset is split into training, validation, and test sets with the ratio of 6:2:2 for four ETT datasets and 7:1:2 for the remaining datasets.

The M4 dataset (Makridakis et al., 2020) consists of 100,000 real-world time series across six frequencies: yearly, quarterly, monthly, weekly, daily, and hourly. It includes data from diverse domains such as finance, economics, demographics, and industry, making it a comprehensive benchmark for evaluating forecasting models. Detailed statistics of the datasets are summarized in Table. 6.

Table 6: Forecasting dataset descriptions. The dataset size is organized in (Train, Validation, Test).

| Tasks | Dataset | Dim | Prediction Length | Dataset Size | Information (Frequency) |
|---|---|---|---|---|---|
| **Long-term** | ETTm1, ETTm2 | 7 | {48, 96, 144, 192} | (34465, 11521, 11521) | Electricity (15 mins) |
| | ETTh1, ETTh2 | 7 | {48, 96, 144, 192} | (8545, 2881, 2881) | Electricity (15 mins) |
| | Electricity | 321 | {48, 96, 144, 192} | (18317, 2633, 5261) | Electricity (Hourly) |
| | Traffic | 862 | {48, 96, 144, 192} | (12185, 1757, 3509) | Transportation (Hourly) |
| | Weather | 21 | {48, 96, 144, 192} | (36792, 5271, 10540) | Weather (10 mins) |
| | ILI | 7 | {24, 36, 48, 60} | (617, 74, 170) | Illness (Weekly) |
| **Short-term** | M4-Yearly | 1 | 6 | (23000, 0, 23000) | Demographic |
| | M4-Quarterly | 1 | 8 | (24000, 0, 24000) | Finance |
| | M4-Monthly | 1 | 18 | (48000, 0, 48000) | Industry |
| | M4-Weekly | 1 | 13 | (359, 0, 359) | Macro |
| | M4-Daily | 1 | 14 | (4227, 0, 4227) | Micro |
| | M4-Hourly | 1 | 48 | (414, 0, 414) | Other |

## A.2. Implementation Details

We implement SKOLR in PyTorch, applying instance-normalization and denormalization (Kim et al., 2022) to inputs and predictions respectively. Following the protocol in the previous works (Zeng et al., 2023; Nie et al., 2023), SKOLR processes each $\mathbf{y}_{1:L,c}$ independently to generate the output $\widehat{\mathbf{y}}_{L+1:L+T,c}$, and subsequently combine them to form a multivariate forecast. This technique is termed *channel-independence* and we omit the variate index $m$ in order to simplify the notation in the Section 3.

To improve computational efficiency, we adopt non-overlapping patch tokenization (Nie et al., 2023) before feeding the frequency-decomposed signals $\{\mathbf{Y}_n\}_{n=1}^N$ into the linear RNN branches. This reduces the sequence length by a factor of $P$, significantly decreasing computation time while maintaining model effectiveness.

The model architecture employs a single-layer linear RNN to preserve linear state transitions between time steps, with

---

[2]https://www.bgc-jena.mpg.de/wetter

[3]https://pems.dot.ca.gov

[4]Wu et al. (2021) selected 321 of 370 customers from the original dataset in Trindade (2015). This version is widely used in the follow-up works.

[5]https://gis.cdc.gov/grasp/fluview/fluportaldashboard.html

MLPs using ReLU activation functions in both encoder and decoder components. The patch length adapts to the input window length as $P = L/6$. Through grid search, we optimize the branch number $N \in \{2, 3, 4, 8\}$, number of MLP layers $M \in \{1, 2, 3\}$ and dynamic dimension $D \in \{128, 256, 512\}$, while maintaining the hidden dimension at $H = 2D$. We train using AdamW optimizer (Loshchilov, 2017) with learning rate $1 \times e^{-4}$ and weight decay $5 \times e^{-4}$, using batch size 32 across all datasets. Complete hyperparameter configurations are detailed in Table 7.

Table 7: Hyperparameters of SKOLR

| Dataset | Traffic | ELC | Weather | ETTh1 | ETTh2 | ETTm1 | ETTm2 | ILI |
|---|---|---|---|---|---|---|---|---|
| Number of Branches $N$ | 2 | 2 | 2 | 2 | 2 | 2 | 2 | 2 |
| Number of MLP hidden layers $M$ | 2 | 2 | 2 | 1 | 1 | 1 | 1 | 1 |
| Dynamic Dimension $D$ | 256 | 256 | 128 | 256 | 128 | 256 | 256 | 256 |
| Dropout | 0.05 | 0.05 | 0.2 | 0.2 | 0.2 | 0.2 | 0.1 | 0.2 |

## B. Additional Experimental Results

### B.1. Short-term Forecasting

#### B.1.1. EXPERIMENTAL SETTING

For the short-term forecasting, following the N-BEATS (Oreshkin et al., 2020), we adopt the symmetric mean absolute percentage error (sMAPE), mean absolute scaled error (MASE), and overall weighted average (OWA) as the metrics, where OWA is a special metric used in the M4 competition. These metrics can be calculated as follows:

$$\text{sMAPE} = \frac{200}{T} \sum_{i=1}^{T} \frac{|Y_i - \hat{Y}_i|}{|Y_i| + |\hat{Y}_i|},$$

$$\text{MASE} = \frac{1}{T} \sum_{i=1}^{T} \frac{|Y_i - \hat{Y}_i|}{\frac{1}{T-m} \sum_{j=m+1}^{T} |Y_j - Y_{j-m}|},$$

$$\text{OWA} = \frac{1}{2} \left( \frac{\text{sMAPE}}{\text{sMAPE}_{\text{Naïve2}}} + \frac{\text{MASE}}{\text{MASE}_{\text{Naïve2}}} \right),$$

where $m$ is the periodicity of the data. $Y, \hat{Y} \in \mathbb{R}^{T \times C}$ are the ground truth and prediction results of the future with $T$ time points and $C$ dimensions. $Y_i$ represents the $i$-th future time point.

We compare SKOLR against state-of-the-art models on the M4 competition dataset, which contains six diverse domains (Yearly, Quarterly, Monthly, Weekly, Daily, Hourly) with prediction horizons ranging from 6 to 48 steps, as shown in Table 6. The baselines includes: N-BEATS (Oreshkin et al., 2020), N-HiTS (Challu et al., 2023), PatchTST (Nie et al., 2023), TimesNet (Wu et al., 2023), DLinear (Zeng et al., 2023), MICN (Wang et al., 2023a), KNF (Wang et al., 2023b), FiLM (Zhou et al., 2022), Autoformer (Wu et al., 2021), and KooPA (Liu et al., 2023). These models are not specifically designed for long-term forecasting, but also generalize well on short-term tasks.

#### B.1.2. RESULTS

Table 8 demonstrates SKOLR's effectiveness in handling diverse temporal patterns across M4 domains. At yearly, quarterly and monthly predictions, where data exhibits strong seasonality and multiple frequency components, SKOLR's parallel branch design allows simultaneous tracking of different temporal scales. Our structured Koopman operator design proves particularly powerful for the M4 dataset, which contains richer dynamic information than typical long-term forecasting benchmarks, resulting in consistently outperforming both traditional forecasting models (N-BEATS, N-HiTS) and other Koopman-based approaches (KooPA) across all evaluation metrics.

Table 8: Comparison of Models for short-term prediction. Best results and second best results are highlighted in **red** and blue respectively.

| M4 | Metric | SKOLR | KooPA | N-HiTS | N-BEATS | PatchTST | TimesNet | DLinear | MICN | KNF | FiLM | Autoformer |
|---|---|---|---|---|---|---|---|---|---|---|---|---|
| Year | sMAPE | **13.291** | 13.352 | 13.371 | 13.466 | 13.517 | 13.394 | 13.866 | 14.532 | 13.986 | 14.012 | 14.786 |
| | MASE | **2.996** | 2.997 | 3.025 | 3.059 | 3.031 | 3.004 | 3.006 | 3.359 | 3.029 | 3.071 | 3.349 |
| | OWA | **0.784** | 0.786 | 0.790 | 0.797 | 0.795 | 0.787 | 0.802 | 0.867 | 0.804 | 0.815 | 0.874 |
| Quarter | sMAPE | **9.986** | 10.159 | 10.454 | 10.074 | 10.847 | 10.101 | 10.689 | 11.395 | 10.343 | 10.758 | 12.125 |
| | MASE | **1.166** | 1.189 | 1.219 | 1.163 | 1.315 | 1.183 | 1.294 | 1.379 | 1.202 | 1.306 | 1.483 |
| | OWA | **0.878** | 0.895 | 0.919 | 0.881 | 0.972 | 0.890 | 0.957 | 1.020 | 0.965 | 0.905 | 1.091 |
| Month | sMAPE | **12.536** | 12.730 | 12.794 | 12.801 | 14.584 | 12.866 | 13.372 | 13.829 | 12.894 | 13.377 | 15.530 |
| | MASE | **0.921** | 0.953 | 0.960 | 0.955 | 1.169 | 0.964 | 1.014 | 1.082 | 1.023 | 1.021 | 1.277 |
| | OWA | **0.867** | 0.901 | 0.895 | 0.893 | 1.055 | 0.894 | 0.940 | 0.988 | 0.985 | 0.944 | 1.139 |
| Others | sMAPE | **4.652** | 4.861 | 4.696 | 5.008 | 6.184 | 4.982 | 4.894 | 6.151 | 4.753 | 5.259 | 5.841 |
| | MASE | 3.233 | **3.124** | 3.130 | 3.443 | 4.818 | 3.323 | 3.358 | 4.263 | 3.138 | 3.608 | 4.308 |
| | OWA | 0.999 | 1.004 | **0.988** | 1.070 | 1.140 | 1.048 | 1.044 | 1.319 | 1.019 | 1.122 | 1.294 |
| Average | sMAPE | **11.704** | 11.863 | 11.960 | 11.910 | 13.022 | 11.930 | 12.418 | 13.023 | 12.126 | 12.489 | 14.057 |
| | MASE | **1.572** | 1.595 | 1.606 | 1.613 | 1.814 | 1.597 | 1.656 | 1.836 | 1.641 | 1.690 | 1.954 |
| | OWA | **0.843** | 0.858 | 0.861 | 0.862 | 0.954 | 0.867 | 0.891 | 0.960 | 0.874 | 0.902 | 1.029 |

Table 9: Prediction results on benchmark datasets with $L = 96$. Best results and second best results are highlighted in **red** and blue respectively.

| Dataset | T | SKOR MSE | SKOR MAE | Koopa MSE | Koopa MAE | iTransformer MSE | iTransformer MAE | PatchTST MSE | PatchTST MAE | Crossformer MSE | Crossformer MAE | TIDE MSE | TIDE MAE | TimesNet MSE | TimesNet MAE | Dlinear MSE | Dlinear MAE | SCINet MSE | SCINet MAE | Stationary MSE | Stationary MAE | Autoformer MSE | Autoformer MAE |
|---|---|---|---|---|---|---|---|---|---|---|---|---|---|---|---|---|---|---|---|---|---|---|---|
| ETTm1 | 96 | **0.313** | **0.356** | 0.330 | 0.368 | 0.334 | 0.368 | 0.329 | 0.367 | 0.404 | 0.426 | 0.364 | 0.387 | 0.338 | 0.375 | 0.345 | 0.372 | 0.418 | 0.438 | 0.386 | 0.398 | 0.505 | 0.475 |
| | 192 | **0.359** | **0.384** | 0.385 | 0.395 | 0.377 | 0.391 | 0.367 | 0.385 | 0.450 | 0.451 | 0.398 | 0.404 | 0.374 | 0.387 | 0.380 | 0.389 | 0.439 | 0.450 | 0.459 | 0.444 | 0.553 | 0.496 |
| | 336 | **0.389** | **0.406** | 0.402 | 0.413 | 0.426 | 0.420 | 0.399 | 0.410 | 0.532 | 0.515 | 0.428 | 0.425 | 0.410 | 0.411 | 0.413 | 0.413 | 0.490 | 0.485 | 0.495 | 0.464 | 0.621 | 0.537 |
| | 720 | **0.449** | 0.443 | 0.472 | 0.449 | 0.491 | 0.459 | 0.454 | 0.439 | 0.666 | 0.589 | 0.487 | 0.461 | 0.478 | 0.450 | 0.474 | 0.453 | 0.595 | 0.550 | 0.585 | 0.516 | 0.671 | 0.561 |
| ETTm2 | 96 | **0.173** | **0.256** | 0.181 | 0.263 | 0.180 | 0.264 | 0.175 | 0.259 | 0.287 | 0.366 | 0.207 | 0.305 | 0.187 | 0.267 | 0.193 | 0.292 | 0.286 | 0.377 | 0.192 | 0.274 | 0.255 | 0.339 |
| | 192 | **0.239** | **0.300** | 0.248 | 0.308 | 0.250 | 0.309 | 0.241 | 0.302 | 0.414 | 0.492 | 0.290 | 0.364 | 0.249 | 0.309 | 0.284 | 0.362 | 0.399 | 0.445 | 0.280 | 0.339 | 0.281 | 0.340 |
| | 336 | **0.302** | **0.341** | 0.303 | 0.343 | 0.311 | 0.348 | 0.305 | 0.343 | 0.597 | 0.542 | 0.377 | 0.422 | 0.321 | 0.351 | 0.369 | 0.427 | 0.637 | 0.591 | 0.334 | 0.361 | 0.339 | 0.372 |
| | 720 | **0.398** | **0.396** | 0.403 | 0.401 | 0.412 | 0.407 | 0.402 | 0.400 | 1.730 | 1.042 | 0.558 | 0.524 | 0.408 | 0.403 | 0.554 | 0.522 | 0.960 | 0.735 | 0.417 | 0.413 | 0.433 | 0.432 |
| ETTh1 | 96 | **0.371** | **0.397** | 0.401 | 0.413 | 0.386 | 0.405 | 0.414 | 0.419 | 0.423 | 0.448 | 0.479 | 0.464 | 0.384 | 0.402 | 0.386 | 0.400 | 0.654 | 0.599 | 0.513 | 0.491 | 0.449 | 0.459 |
| | 192 | **0.423** | **0.427** | 0.449 | 0.439 | 0.441 | 0.436 | 0.460 | 0.445 | 0.471 | 0.474 | 0.525 | 0.492 | 0.436 | 0.429 | 0.437 | 0.432 | 0.719 | 0.631 | 0.534 | 0.504 | 0.500 | 0.482 |
| | 336 | **0.471** | **0.453** | 0.494 | 0.461 | 0.487 | 0.458 | 0.501 | 0.466 | 0.570 | 0.546 | 0.565 | 0.515 | 0.491 | 0.469 | 0.481 | 0.459 | 0.778 | 0.659 | 0.588 | 0.535 | 0.521 | 0.496 |
| | 720 | 0.499 | 0.484 | **0.484** | **0.472** | 0.503 | 0.491 | 0.500 | 0.488 | 0.653 | 0.621 | 0.594 | 0.558 | 0.521 | 0.500 | 0.519 | 0.516 | 0.836 | 0.699 | 0.643 | 0.616 | 0.514 | 0.512 |
| ETTh2 | 96 | **0.293** | **0.344** | 0.316 | 0.361 | 0.297 | 0.349 | 0.302 | 0.348 | 0.745 | 0.584 | 0.400 | 0.440 | 0.340 | 0.374 | 0.333 | 0.387 | 0.707 | 0.621 | 0.476 | 0.458 | 0.346 | 0.388 |
| | 192 | **0.370** | **0.384** | 0.384 | 0.405 | 0.380 | 0.400 | 0.388 | 0.400 | 0.877 | 0.656 | 0.528 | 0.509 | 0.402 | 0.414 | 0.477 | 0.476 | 0.860 | 0.689 | 0.512 | 0.493 | 0.456 | 0.452 |
| | 336 | **0.410** | **0.428** | 0.423 | 0.438 | 0.428 | 0.432 | 0.426 | 0.433 | 1.043 | 0.731 | 0.643 | 0.571 | 0.452 | 0.452 | 0.594 | 0.541 | 1.000 | 0.744 | 0.552 | 0.551 | 0.482 | 0.486 |
| | 720 | 0.431 | 0.446 | 0.450 | 0.458 | **0.427** | **0.445** | 0.431 | 0.446 | 1.104 | 0.763 | 0.874 | 0.679 | 0.462 | 0.468 | 0.831 | 0.657 | 1.249 | 0.838 | 0.562 | 0.560 | 0.515 | 0.511 |
| ECL | 96 | 0.153 | 0.246 | **0.146** | 0.244 | 0.148 | **0.240** | 0.181 | 0.270 | 0.219 | 0.314 | 0.237 | 0.329 | 0.168 | 0.272 | 0.197 | 0.282 | 0.247 | 0.345 | 0.169 | 0.273 | 0.201 | 0.317 |
| | 192 | 0.168 | 0.259 | 0.169 | 0.266 | **0.162** | **0.253** | 0.188 | 0.274 | 0.231 | 0.322 | 0.236 | 0.330 | 0.184 | 0.289 | 0.196 | 0.285 | 0.257 | 0.355 | 0.182 | 0.286 | 0.222 | 0.334 |
| | 336 | 0.189 | 0.282 | 0.189 | 0.285 | **0.178** | **0.269** | 0.204 | 0.293 | 0.246 | 0.337 | 0.249 | 0.344 | 0.198 | 0.300 | 0.209 | 0.301 | 0.269 | 0.369 | 0.200 | 0.304 | 0.231 | 0.338 |
| | 720 | 0.230 | 0.318 | 0.226 | **0.314** | 0.225 | 0.317 | 0.246 | 0.324 | 0.280 | 0.363 | 0.284 | 0.373 | **0.220** | 0.320 | 0.245 | 0.333 | 0.299 | 0.390 | 0.222 | 0.321 | 0.254 | 0.361 |
| Traffic | 96 | 0.427 | 0.270 | 0.462 | 0.290 | **0.395** | **0.268** | 0.462 | 0.295 | 0.522 | 0.290 | 0.805 | 0.493 | 0.593 | 0.321 | 0.650 | 0.396 | 0.788 | 0.499 | 0.612 | 0.338 | 0.613 | 0.388 |
| | 192 | 0.455 | 0.289 | 0.566 | 0.386 | **0.417** | **0.276** | 0.466 | 0.296 | 0.530 | 0.293 | 0.756 | 0.474 | 0.617 | 0.336 | 0.598 | 0.370 | 0.789 | 0.505 | 0.613 | 0.340 | 0.616 | 0.382 |
| | 336 | 0.472 | 0.298 | 0.514 | 0.331 | **0.433** | **0.283** | 0.482 | 0.304 | 0.558 | 0.305 | 0.762 | 0.477 | 0.629 | 0.336 | 0.605 | 0.373 | 0.797 | 0.508 | 0.618 | 0.328 | 0.622 | 0.337 |
| | 720 | 0.519 | 0.326 | 0.552 | 0.346 | **0.467** | **0.302** | 0.514 | 0.322 | 0.589 | 0.328 | 0.719 | 0.449 | 0.640 | 0.350 | 0.645 | 0.394 | 0.841 | 0.523 | 0.653 | 0.355 | 0.660 | 0.408 |
| Weather | 96 | 0.162 | 0.207 | **0.157** | **0.202** | 0.174 | 0.214 | 0.177 | 0.218 | 0.158 | 0.230 | 0.202 | 0.261 | 0.172 | 0.220 | 0.196 | 0.255 | 0.221 | 0.306 | 0.173 | 0.223 | 0.266 | 0.336 |
| | 192 | 0.208 | 0.249 | 0.209 | 0.251 | 0.221 | 0.254 | 0.225 | 0.259 | **0.206** | 0.277 | 0.242 | 0.298 | 0.219 | 0.261 | 0.237 | 0.296 | 0.261 | 0.340 | 0.245 | 0.285 | 0.307 | 0.367 |
| | 336 | **0.266** | 0.292 | 0.266 | **0.290** | 0.278 | 0.296 | 0.278 | 0.297 | 0.272 | 0.335 | 0.287 | 0.335 | 0.280 | 0.306 | 0.283 | 0.335 | 0.309 | 0.378 | 0.321 | 0.338 | 0.359 | 0.395 |
| | 720 | **0.344** | **0.343** | 0.350 | 0.348 | 0.358 | 0.347 | 0.354 | 0.348 | 0.398 | 0.418 | 0.351 | 0.386 | 0.365 | 0.359 | 0.345 | 0.381 | 0.377 | 0.427 | 0.414 | 0.410 | 0.419 | 0.428 |

## B.2. Results for a shorter look-back window

For a fair comparison, we also conduct the experiments under the $L = 96$ setting that is the default for iTransformer, TimesNet, and other transformer-based models, except PatchTST. For the shorter look-back window, we use the patch length $P = 16$ to obtain the tokens for SKOLR. By default, SKOLR follow the hyperparameters in Table 7. As shown in Table 9, SKOLR emerges as the leading performer, achieving Rank 1 or 2 in 24 cases out of 28 cases in terms of MSE and 25 cases in terms of MAE.

Table 10: Model performance across different datasets with mean $\pm$ standard deviation for MSE and MAE metrics.

| Dataset | Models | MSE | MAE |
|---------|--------|-----|-----|
| ECL | 48 | $0.137 \pm 0.0003$ | $0.229 \pm 0.0003$ |
| | 96 | $0.132 \pm 0.0005$ | $0.225 \pm 0.0004$ |
| | 144 | $0.143 \pm 0.0001$ | $0.236 \pm 0.0001$ |
| | 192 | $0.149 \pm 0.0001$ | $0.244 \pm 0.0001$ |
| Traffic | 48 | $0.400 \pm 0.0003$ | $0.258 \pm 0.0040$ |
| | 96 | $0.368 \pm 0.0007$ | $0.248 \pm 0.0007$ |
| | 144 | $0.375 \pm 0.0003$ | $0.255 \pm 0.0002$ |
| | 192 | $0.377 \pm 0.0003$ | $0.256 \pm 0.0002$ |
| Weather | 48 | $0.131 \pm 0.0009$ | $0.170 \pm 0.0008$ |
| | 96 | $0.154 \pm 0.0015$ | $0.202 \pm 0.0015$ |
| | 144 | $0.172 \pm 0.0009$ | $0.220 \pm 0.0006$ |
| | 192 | $0.193 \pm 0.0004$ | $0.241 \pm 0.0005$ |
| ETTm1 | 48 | $0.280 \pm 0.0013$ | $0.330 \pm 0.0015$ |
| | 96 | $0.287 \pm 0.0003$ | $0.340 \pm 0.0001$ |
| | 144 | $0.313 \pm 0.0020$ | $0.361 \pm 0.0023$ |
| | 192 | $0.328 \pm 0.0019$ | $0.373 \pm 0.0018$ |
| ETTm2 | 48 | $0.134 \pm 0.0011$ | $0.228 \pm 0.0007$ |
| | 96 | $0.171 \pm 0.0015$ | $0.255 \pm 0.0013$ |
| | 144 | $0.209 \pm 0.0014$ | $0.283 \pm 0.0014$ |
| | 192 | $0.241 \pm 0.0013$ | $0.304 \pm 0.0015$ |
| ETTh1 | 48 | $0.333 \pm 0.0009$ | $0.373 \pm 0.0007$ |
| | 96 | $0.371 \pm 0.0011$ | $0.398 \pm 0.0008$ |
| | 144 | $0.405 \pm 0.0019$ | $0.417 \pm 0.0020$ |
| | 192 | $0.422 \pm 0.0030$ | $0.432 \pm 0.0034$ |
| ETTh2 | 48 | $0.238 \pm 0.0012$ | $0.306 \pm 0.0004$ |
| | 96 | $0.299 \pm 0.0034$ | $0.352 \pm 0.0042$ |
| | 144 | $0.335 \pm 0.0042$ | $0.377 \pm 0.0048$ |
| | 192 | $0.365 \pm 0.0033$ | $0.397 \pm 0.0040$ |
| ILI | 24 | $1.556 \pm 0.0213$ | $0.760 \pm 0.0159$ |
| | 36 | $1.462 \pm 0.0711$ | $0.728 \pm 0.0676$ |
| | 48 | $1.537 \pm 0.0038$ | $0.798 \pm 0.0030$ |
| | 60 | $2.187 \pm 0.0435$ | $0.995 \pm 0.0498$ |

### B.3. Experimental Variability

Table 10 reports standard deviation (std) across 3 independent runs for all datasets and forecast horizons. The low stds (<0.003 for most datasets) demonstrate the consistency of SKOLR's performance.

### B.4. Comparison with Orvieto et al. (2023)

The Linear Recurrent Unit (LRU) presented by Orvieto et al. (2023) is derived from vanilla recurrent neural networks (RNNs) through a sequence of principled modifications including linearization of the recurrence, diagonalization, exponential parametrization for stability, and forward-pass normalization. These changes yield a model that can match the performance of recent deep state-space models (SSMs) such as S4 and S5 on benchmarks like the Long Range Arena (Tay et al., 2021), without relying on discretization of continuous-time dynamics.

We implement our code in PyTorch. Our implementation follows from the JAX pseudocode presented in the original paper's appendix (Orvieto et al., 2023). Additionally, we consulted a community implementation in PyTorch[6]. The LRU is trained using the AdamW optimizer with no weight decay applied to the recurrent parameters. Learning rates are selected via grid search on a logarithmic scale. All experiments use networks of 6 LRU layers with residual and normalization layers between blocks and a final linear output layer and a 64-dimensional hidden state.

---

[6]https://github.com/Gothos/LRU-pytorch

Table 11: Performance comparison of LRU, Koopa and SKOLR on non-linear dynamical systems (NLDS)

| | SKOLR | | KooPA | | LRU | |
|---|---|---|---|---|---|---|
| **Dataset** | MSE | MAE | MSE | MAE | MSE | MAE |
| Pendulum | **0.0001** | **0.0083** | 0.0039 | 0.0470 | 0.0572 | 0.0242 |
| Duffing | **0.0047** | **0.0518** | 0.0365 | 0.1479 | 0.0573 | 0.5970 |
| Lotka-Volterra | **0.0018** | **0.0354** | 0.0178 | 0.1050 | 0.2058 | 0.3779 |
| Lorenz '63 | **0.9740** | **0.7941** | 1.0937 | 0.8325 | 1.1905 | 0.8932 |

Whereas our focus in SKOLR is time-series forecasting, Orvieto et al. (2023) target long-range reasoning. Although it is possible to convert their architecture to address forecasting, performance suffers because it is not the design goal, as shown in Table 11.

## C. Model Efficiency

### C.1. Theoretical Complexity Analysis

SKOLR achieves computational efficiency through its structured design and linear operations. For a time series of length $L$ with patch length $P$, embedding dimension $D$, and $N$ branches, we analyze both time and space complexity.

The time complexity of SKOLR consists of several components: spectral decomposition, encoder/decoder MLPs, and linear RNN computation. If we perform a single FFT operation $O(L \log L)$ followed by branch-specific frequency filtering, the main computational cost comes from encoder/decoder MLPs $O(N \times (L/P) \times D^2)$ and linear RNN computation $O(N \times (L/P) \times D^2)$, giving a total time complexity of $O(N \times (L/P) \times D^2)$. The memory complexity includes model parameters $O(N \times D^2)$ and activation memory $O(N \times (L/P) \times D)$.

Our structured approach with $N$ branches provides substantial efficiency gains compared to a non-structured approach with equivalent representational capacity. For a non-structured model with dimension $D' = N \times D$, the time complexity would be $O((L/P) \times N^2 D^2)$ and memory complexity $O(N^2 D^2)$. This represents an $N$-fold increase in computational requirements. For example, with $N = 16$ branches, our structured approach requires approximately $16\times$ fewer parameters and operations while maintaining equivalent or better modeling capacity, as shown in Section 4.3.1. Compared to transformer-based approaches with time complexity $O((L/P)^2 \times D + (L/P) \times D^2)$ and memory complexity $O((L/P)^2 + (L/P) \times D)$, SKOLR demonstrates a fundamental advantage for long sequences by avoiding the quadratic scaling with sequence length.

### C.2. Parallel Computing

SKOLR further benefits from parallel processing capabilities. The $N$ separate branches can be processed completely independently, reducing the effective time complexity to $O((L/P) \times D^2)$ with sufficient parallel resources. This branch-level parallelism is implemented in our current code.

In future work, we plan to implement additional parallelization of the linear RNN computation itself. Since our RNN has no activation functions, we can express the hidden state evolution for each branch with sequence length $L/P$ in closed form: $h_k = g(y_k) + \sum_{s=1}^{L/P} W^s g(y_{k-s})$, where $W^s$ indicates $s$ applications of $W$ (the state transition matrix). This formulation allows us to compute all hidden states simultaneously through efficient matrix operations, potentially reducing the time complexity further to $O(D^3 \log(L/P) + (L/P)^2 \times D)$ per branch.

For time series with patching where $L/P \ll D$, this approach achieves significant speedups by eliminating the sequential dependency in RNN computation. With both branch and RNN parallelism implemented, SKOLR can achieve greater computational efficiency while maintaining its forecasting performance.

### C.3. Computational Efficiency

We provided efficiency results on the ETTm2 and Traffic datasets in Fig. 5. We have expanded our evaluation across additional datasets to offer a more comprehensive assessment in Table 12. In all datasets, the proposed architecture provides a compelling trade-off between efficiency and accuracy compared to baselines.

Table 12: Model Efficiency and Performance Comparison for Different Datasets with $T = 96$. Parameters (Params) are measured in millions (M), GPU memory (GPU) in MiB, computation time per epoch in seconds (s) on NVIDIA V100 GPU with batch size 32.

(a) Traffic

| Model | Params (M) | GPU(MiB) | Time (s) | MSE |
|---|---|---|---|---|
| Autoformer | 14.914 | 18.811 | 51.0 | 0.668 |
| iTransformer | 6.405 | 62.710 | 126.0 | 0.388 |
| PatchTST | 3.755 | 22.132 | 1042.0 | 0.413 |
| MICN | 236.151 | 32.310 | 84.0 | 0.511 |
| TimesNet | 30.170 | 111.998 | 6563.0 | 0.611 |
| DLinear | 0.009 | 12.861 | 7.7 | 0.485 |
| Koopa | 5.429 | 50.335 | 25.5 | 0.401 |
| SKOLR | 1.479 | 5.915 | 216.0 | 0.368 |

(b) Electricity

| Model | Params (M) | GPU(MiB) | Time (s) | MSE |
|---|---|---|---|---|
| Autoformer | 11.214 | 17.373 | 68.7 | 0.182 |
| iTransformer | 4.957 | 86.478 | 58.6 | 0.134 |
| PatchTST | 6.904 | 73.517 | 1231.0 | 0.143 |
| MICN | 6.635 | 32.668 | 18.0 | 0.165 |
| TimesNet | 15.037 | 33.435 | 11351.0 | 0.170 |
| DLinear | 0.019 | 76.016 | 6.8 | 0.153 |
| Koopa | 4.076 | 31.067 | 33.1 | 0.136 |
| SKOLR | 1.541 | 6.163 | 99.1 | 0.132 |

(c) ETTh1

| Model | Params (M) | GPU(MiB) | Time (s) | MSE |
|---|---|---|---|---|
| Autoformer | 10.536 | 16.523 | 29.5 | 0.634 |
| iTransformer | 0.237 | 27.245 | 4.1 | 0.393 |
| PatchTST | 3.752 | 22.018 | 8.5 | 0.372 |
| MICN | 252.001 | 65.974 | 21.1 | 0.406 |
| TimesNet | 0.605 | 26.053 | 22.1 | 0.411 |
| DLinear | 0.140 | 26.440 | 0.6 | 0.379 |
| Koopa | 0.135 | 31.951 | 10.1 | 0.371 |
| SKOLR | 0.429 | 1.717 | 2.8 | 0.371 |

(d) ETTm2

| Model | Params (M) | GPU(MiB) | Time (s) | MSE |
|---|---|---|---|---|
| Autoformer | 10.536 | 14.599 | 152.6 | 0.241 |
| iTransformer | 0.237 | 27.245 | 13.1 | 0.177 |
| PatchTST | 10.056 | 39.910 | 980.0 | 0.171 |
| MICN | 252.001 | 65.974 | 84.2 | 0.197 |
| TimesNet | 1.192 | 34.783 | 113.0 | 0.187 |
| DLinear | 18.291 | 9.312 | 1.9 | 0.172 |
| Koopa | 0.135 | 31.951 | 48.2 | 0.171 |
| SKOLR | 0.429 | 1.717 | 12.6 | 0.171 |

# D. Additional Analysis

## D.1. Scaling Up Forecast Horizon

We have conducted experiments to explore performance in the setting where the forecast horizon is increased at test-time. In this experiment, SKOLR and Koopa were evaluated by scaling up from the training horizon ($T_{tr}$) to a larger test horizon ($T_{te}$). Unlike Koopa (Liu et al., 2023), SKOLR does not incorporate an operator adaptation (OA) mechanism to update its Koopman operator using incoming ground truth. Instead, our architecture possesses a natural recursive structure that enables straightforward extension to longer horizons. Even when weights are trained to minimize a loss function specified over a given horizon, the algorithm can be recursively applied to predict over extended periods.

As demonstrated in Table 13, SKOLR maintains competitive performance without requiring additional adaptation mechanisms. The structured Koopman operator and linear RNN design enable robust long-term predictions, with error percentages remaining comparable to Koopa OA across various datasets. This demonstrates SKOLR's inherent capability to handle extended forecast horizons efficiently through its recursive architecture.

## D.2. Ablation Study

We have also conducted a more comprehensive ablation study on the design elements of SKOLR. As shown in Table 14, we compare our full SKOLR model with two ablated variants: (1) "w/o Structure": no structured decomposition, using a single branch with dimension (N×D); (2) "w/o Spectral Encoder": no learnable frequency decomposition, while maintaining the multi-branch structure.

The results show that both components contribute meaningfully. Removing the structured decomposition leads to performance degradation on 27/32 tasks, with notable declines on ETTh1 and ILI, while increasing computational overhead. Similarly, removing the spectral encoder impacts performance on 23/32 tasks, though with a smaller overall effect.

Table 13: Scaling up forecast horizon: (T_tr, T_te) = (24, 48) for ILI and (T_tr, T_te) = (48, 144) for others. Koopa and SKOLR conducts vanilla rolling forecast and Koopa OA has operator adaptation.

| | ETTh2 (ADF -4.135) | | ILI (ADF -5.406) | | ECL (ADF -8.483) | | Traffic (ADF -15.046) | | Weather (ADF -26.661) | |
|---|---|---|---|---|---|---|---|---|---|---|
| Metric | MSE | MAE | MSE | MAE | MSE | MAE | MSE | MAE | MSE | MAE |
| Koopa (T_tr) | 0.226 | 0.300 | 1.621 | 0.800 | 0.130 | 0.234 | 0.415 | 0.274 | 0.126 | 0.168 |
| **Koopa(T_te)** | 0.437 | 0.429 | 2.836 | 1.065 | 0.199 | 0.298 | 0.709 | 0.437 | 0.237 | 0.276 |
| Error(+ %) | 93% | 43% | 75% | 33% | 53% | 27% | 71% | 59% | 88% | 64% |
| **Koopa OA(T_te)** | 0.372 | 0.404 | 2.427 | 0.907 | 0.182 | 0.271 | 0.699 | 0.426 | 0.225 | 0.264 |
| Error(+ %) | 65% | 35% | 50% | 13% | 40% | 16% | 68% | 55% | 79% | 57% |
| SKOLR (T_tr) | 0.238 | 0.306 | 1.556 | 0.760 | 0.137 | 0.229 | 0.400 | 0.258 | 0.131 | 0.170 |
| **SKOLR (T_te)** | 0.393 | 0.402 | 2.392 | 0.958 | 0.204 | 0.289 | 0.612 | 0.383 | 0.222 | 0.257 |
| Error(+ %) | 65% | 31% | 54% | 26% | 49% | 26% | 53% | 48% | 69% | 51% |

Table 14: Ablation study comparing SKOLR with versions without structure and without spectral encoder

| Dataset | T | SKOLR | | w/o Structure | | w/o Spectral Encoder | |
|---|---|---|---|---|---|---|---|
| | | MSE | MAE | MSE | MAE | MSE | MAE |
| ECL | 48 | **0.137** | **0.229** | _0.148_ | _0.238_ | 0.149 | 0.238 |
| | 96 | **0.132** | **0.225** | 0.135 | 0.228 | _0.133_ | _0.227_ |
| | 144 | _0.143_ | _0.236_ | 0.146 | 0.241 | **0.142** | **0.235** |
| | 192 | _0.149_ | _0.244_ | 0.150 | 0.245 | **0.148** | **0.243** |
| Traffic | 48 | 0.400 | 0.258 | **0.395** | **0.255** | _0.397_ | _0.257_ |
| | 96 | _0.368_ | **0.248** | **0.367** | _0.249_ | 0.369 | _0.249_ |
| | 144 | 0.375 | 0.255 | 0.375 | 0.255 | 0.375 | 0.255 |
| | 192 | **0.377** | 0.256 | 0.378 | 0.256 | **0.377** | 0.256 |
| Weather | 48 | **0.131** | **0.170** | _0.134_ | 0.173 | 0.134 | _0.172_ |
| | 96 | **0.154** | **0.202** | _0.157_ | 0.203 | 0.158 | **0.202** |
| | 144 | **0.172** | **0.220** | 0.177 | 0.225 | _0.175_ | _0.221_ |
| | 192 | **0.193** | **0.241** | 0.195 | 0.242 | 0.197 | 0.244 |
| ETTm1 | 48 | **0.280** | **0.330** | 0.284 | 0.334 | _0.282_ | _0.332_ |
| | 96 | **0.287** | **0.340** | 0.292 | 0.343 | _0.291_ | _0.342_ |
| | 144 | **0.313** | **0.361** | 0.325 | 0.365 | _0.319_ | **0.361** |
| | 192 | **0.328** | 0.373 | 0.332 | **0.372** | 0.332 | **0.372** |
| ETTm2 | 48 | **0.134** | **0.228** | _0.135_ | _0.229_ | 0.162 | 0.259 |
| | 96 | _0.171_ | _0.255_ | 0.174 | 0.259 | **0.169** | **0.253** |
| | 144 | _0.209_ | 0.283 | **0.206** | **0.280** | _0.209_ | _0.282_ |
| | 192 | 0.241 | _0.304_ | 0.241 | 0.305 | **0.230** | **0.299** |
| ETTh1 | 48 | **0.333** | **0.373** | 0.338 | 0.377 | _0.336_ | _0.374_ |
| | 96 | **0.371** | **0.398** | 0.387 | 0.408 | _0.373_ | _0.399_ |
| | 144 | **0.405** | **0.417** | 0.414 | 0.423 | _0.410_ | _0.420_ |
| | 192 | 0.422 | 0.432 | **0.409** | **0.421** | _0.413_ | _0.422_ |
| ETTh2 | 48 | _0.238_ | 0.306 | **0.233** | **0.304** | 0.239 | _0.305_ |
| | 96 | **0.299** | **0.352** | _0.301_ | _0.350_ | 0.303 | 0.350 |
| | 144 | **0.335** | **0.377** | 0.341 | 0.382 | _0.337_ | _0.381_ |
| | 192 | **0.365** | **0.397** | 0.370 | _0.398_ | 0.370 | 0.401 |
| ILI | 24 | 1.556 | 0.760 | 1.795 | 0.842 | **1.522** | **0.741** |
| | 36 | **1.462** | **0.728** | 1.990 | 0.889 | _1.496_ | _0.734_ |
| | 48 | **1.537** | **0.798** | 1.875 | 0.909 | _1.571_ | _0.810_ |
| | 60 | **2.187** | **0.995** | 2.407 | 1.056 | _2.263_ | _0.999_ |

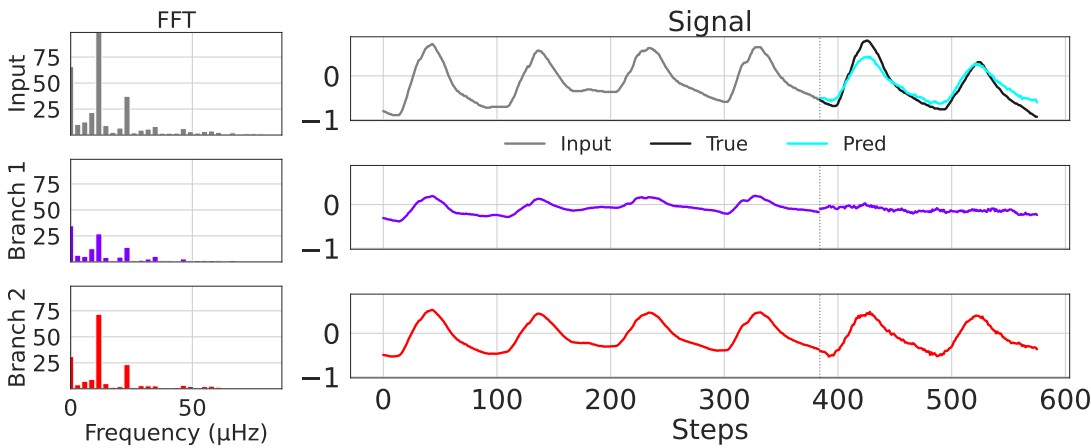

Figure 6: Analysis of SKOLR's branch-wise behavior on ETTm2 feature 6: (a) frequency decomposition and (b) prediction performance.

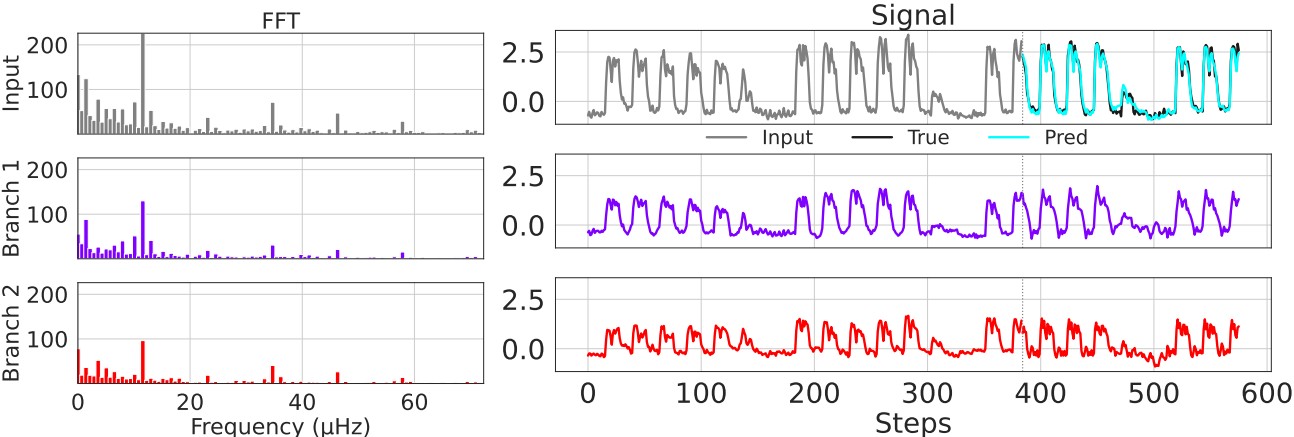

Figure 7: Analysis of SKOLR's branch-wise behavior on Electricity feature 11: (a) frequency decomposition and (b) prediction performance.

### D.3. Branch-wise Visualization

We add examples from ETTm2 and Electricity in order to analyze SKOLR's branch-wise behavior. The figures show distinct frequency specializations. In Fig. 6, SKOLR decomposes the time series into complementary frequency components, with Branch 1 focusing on lower, distributed frequencies, and Branch 2 capturing more specific dominant frequency peaks. In Fig. 7, Branch 1 shows higher amplitudes across most of the very low ($0 - 20\mu Hz$) frequency components compared to Branch 2. The time-domain plots demonstrate how these spectral differences translate into signal reconstruction; Branch 2 focuses more on prediction of the higher-frequency components of the time series.

### D.4. Analysis of Error Accumulation

Time-series forecasting has two main prediction methods: direct prediction, which forecasts the entire horizon at once but is parameter-inefficient and cannot extend the prediction horizon after training, and recursive prediction, which iteratively uses predictions as inputs but may suffer from error accumulation over long sequences.

In SKOLR, we use a patching approach (Appendix A.2) to create an effective middle ground between these methods. Instead of operating at the individual timestep level, we work with patches of multiple timesteps, directly predicting all values within each patch while only applying recursion between patches. This dramatically reduces the number of recursive steps (e.g., from 720 to just 5 with patch length 144), controlling error accumulation. Additionally, this method also reduces complexity to $O(\frac{L}{P})$) from RNN standard timestamp-based approaches ($O(L)$) while maintaining the core principle of Koopman operator theory.

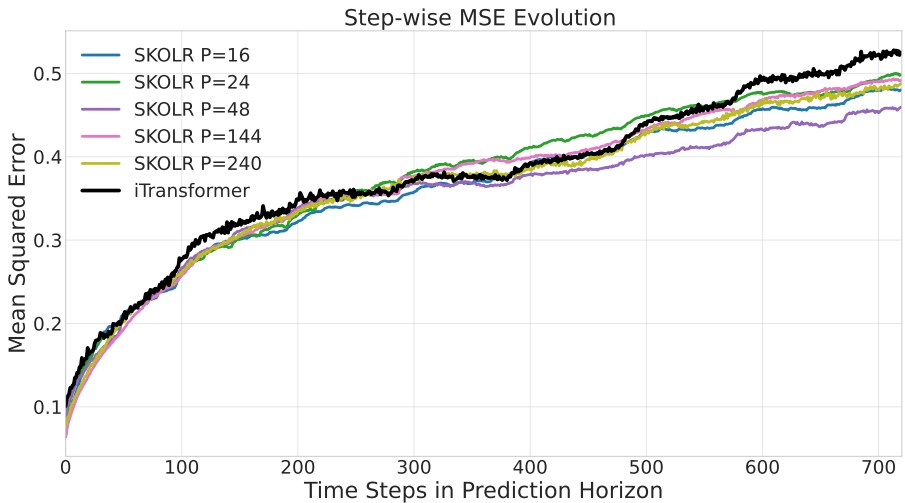

Figure 8: Error progression across 720 time steps: SKOLR (multiple patch sizes) vs. iTransformer

To better address this question, we add an experiment on a longer horizon $L = 720, T = 720$ on dataset ETTm2 with varying iteration steps. We vary the patch length $P = \{16, 24, 48, 144, 240\}$ of the SKOLR model to see the difference performance caused by the number of iterations. SKOLR with $P = 16$ requires 45 recursive steps, while $P = 240$ needs only 3, yet they maintain comparable error profiles in Fig. 8. This empirically demonstrates that SKOLR's patch-based approach effectively controls error accumulation, even with increased recursion.

We also compare SKOLR with iTransformer (Liu et al., 2024), which performs direct prediction without recurrence. Both models show similar patterns of error increase with longer horizons, suggesting that this modest increase is inherent to all forecasting approaches when extending the prediction range, rather than being caused by recurrent error accumulation.

### D.5. Analysis on Koopman operator eigenvalue

The Koopman modes are derived through eigendecomposition of the RNN weight matrices $M_i$. These modes represent dynamical patterns in the data. Each mode captures specific components of the time series. The stability and oscillatory behavior of each mode is determined by the corresponding eigenvalue's position in the complex plane. The eigenvalue plots in Fig. 9 show that each branch learns complementary spectral properties, with all eigenvalues within the unit circle, indicating stable dynamics. Branch 1 shows concentration at magnitude 0.4, while Branch 2 exhibits a more uniform distribution.

Moreover, this observation provides some reassurance against error accumulation concerns, as error divergence is more likely for unstable systems. Our learned system's stability encourages error effects to naturally decay over time during forward prediction steps rather than compounding.

## E. State Prediction for Nonlinear Dynamical Systems (NLDS)

We generated datasets for four nonlinear dynamical systems (NLDS) to evaluate the performance of Koopman-based models in state prediction tasks. Each dataset contains a trajectory with a total of 20000 time steps. The first 14000 steps were designated for training, 2000 steps were used for validation, while the remaining 4,000 steps were used for inference. Below, we describe the generation process for each system:

- **Pendulum:** A simple nonlinear pendulum system described by its angular displacement and velocity. The trajectory was initialized with random a angle and angular velocity, with updates computed using the equations of motion under gravity. The system being simulated is a simple pendulum, consisting of a mass $m$ attached to the end of a rigid, massless rod of length $l$. The pendulum swings in a two-dimensional plane under the influence of gravity, with the gravitational acceleration $g$. The motion of the pendulum is governed by the equation of motion:

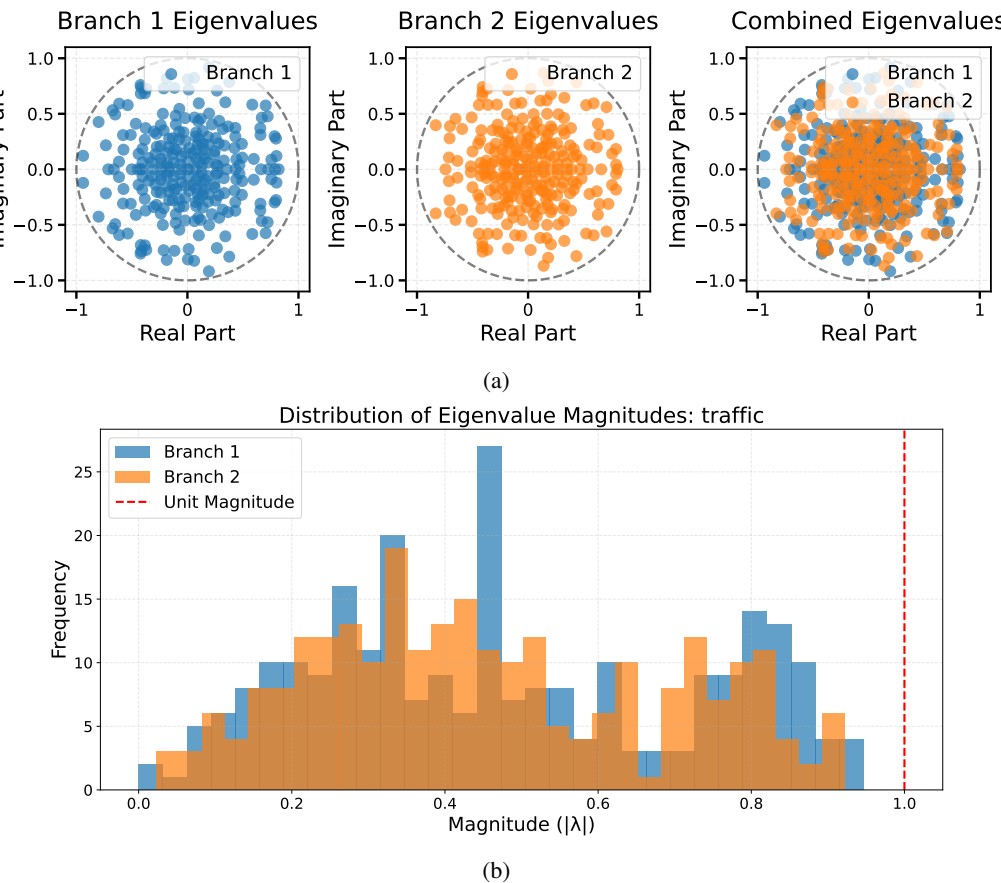

Figure 9: Koopman operator eigenvalue analysis for SKOLR on the Traffic dataset

$$\theta''(t) + \frac{g}{l}\sin(\theta(t)) = 0 \tag{17}$$

where:

- $\theta(t)$ is the angular displacement of the pendulum at time $t$,
- $\theta''(t)$ is the angular acceleration,
- $g$ is the acceleration due to gravity ($9.81\,\text{m/s}^2$),
- $l$ is the length of the pendulum (set to $1.0\,\text{m}$).

The system is further characterized by its initial conditions:

- The initial angle $\theta_0$, which is randomly chosen from a uniform distribution between $-\pi$ and $\pi$,
- The initial angular velocity $\omega_0$, which is randomly chosen from a uniform distribution between $-1\,\text{rad/s}$ and $1\,\text{rad/s}$.

The motion of the pendulum is modeled using numerical methods, specifically the Euler method, which approximates the solution of the system of equations over discrete time steps.

- **Duffing Oscillator:** A nonlinear oscillator characterized by damping and cubic stiffness terms. Trajectories were generated using randomized initial positions and velocities, with dynamics influenced by an external periodic driving force. The system modeled by the code is a Duffing oscillator, a type of nonlinear second-order differential equation

commonly used to describe systems with nonlinear restoring forces and damping. The equation of motion for the Duffing oscillator is given by:

$$\ddot{x} + \delta\dot{x} + \alpha x + \beta x^3 = \gamma\cos(\omega t)$$

where $x(t)$ represents the displacement of the oscillator, $y(t) = \dot{x}(t)$ represents its velocity, and $t$ is time. The parameters of the system are: $\alpha = 1.0$ (linear stiffness), $\beta = 5.0$ (nonlinear stiffness), $\delta = 0.3$ (damping coefficient), $\gamma = 8.0$ (driving force amplitude), and $\omega = 0.5$ (angular frequency of the driving force). The system undergoes periodic driving forces, and its motion is influenced by both the nonlinear restoring force and damping. The motion of the oscillator is simulated by numerically integrating the equations of motion using a simple time-stepping method, where dt is the time step, and the initial conditions for $x$ and $y$ are randomly selected within a small range. The system's behavior is characterized by chaotic dynamics for the chosen parameter values.

- **Lotka-Volterra:** A predator-prey population model, where the prey and predator populations interact dynamically. Trajectories were initialized with random population sizes, and updates followed the Lotka-Volterra equations. The equations governing the Lotka-Volterra predator-prey model are given by:

$$\frac{dN_{\text{prey}}}{dt} = \alpha N_{\text{prey}} - \beta N_{\text{prey}} N_{\text{predator}}$$

$$\frac{dN_{\text{predator}}}{dt} = \delta N_{\text{prey}} N_{\text{predator}} - \gamma N_{\text{predator}}$$

where:

- $N_{\text{prey}}$ is the population of the prey species,
- $N_{\text{predator}}$ is the population of the predator species,
- $\alpha$ is the natural growth rate of the prey,
- $\beta$ is the predation rate (rate at which predators kill prey),
- $\delta$ is the rate at which predators increase due to consuming prey,
- $\gamma$ is the natural death rate of the predator.

In this model, the prey species grows exponentially in the absence of predators, and the predator species declines exponentially in the absence of prey. The interaction between the species causes cyclical fluctuations in their populations.

We implement this model by numerically integrating the differential equations using a simple Euler method. The process starts by initializing the prey and predator populations randomly within a given range. The parameters $\alpha = 1.1$, $\beta = 0.4$, $\delta = 0.1$, and $\gamma = 0.4$ are then used to update the populations at each time step.

- **Lorenz '63:** A chaotic system described by three variables: $x$, $y$, and $z$. Each trajectory is started with randomized initial conditions, and updated using the Lorenz equations with standard parameters. The equations governing the Lorenz system are given by:

$$\frac{dx}{dt} = \sigma(y - x)$$

$$\frac{dy}{dt} = x(\rho - z) - y$$

$$\frac{dz}{dt} = xy - \beta z$$

where:

- $x$, $y$, and $z$ represent the state variables of the system, typically interpreted as the variables describing the convection rolls in the atmosphere,
- $\sigma$ is the Prandtl number, a measure of the fluid's viscosity, set to 10.0,

- $\rho$ is the Rayleigh number, representing the temperature difference between the top and bottom of the fluid, set to 28.0,
- $\beta$ is a geometric factor, set to $\frac{8}{3}$.

The Lorenz system exhibits chaotic behavior for these parameter values, meaning that small differences in initial conditions can lead to vastly different outcomes over time. In the simulation, the system of differential equations is solved using the Euler method over a series of time steps. A visualization of the system is shown in Fig. 10.

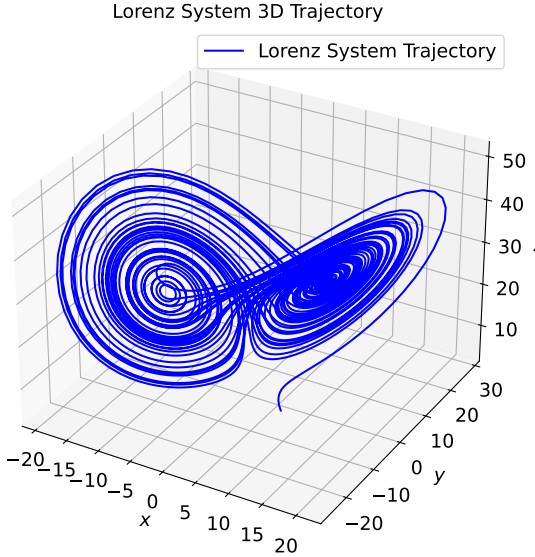

Figure 10: Lorentz '63 system plotted in 3D

