# OpenReview forum: "SKOLR: Structured Koopman Operator Linear RNN for Time-Series Forecasting"
_ICML.cc/2025/Conference — ICML 2025 poster_

### Official Review · Reviewer_qBxE · 2025-03-11

**Overall Recommendation:** 1

**Summary:**

This paper introduces SKOLR, a structured Koopman operator-based approach to time-series forecasting that connects Koopman operator approximations with linear recurrent neural networks (RNNs). By representing dynamic states using time-delayed observations, the method approximates the Koopman operator in a structured manner through a parallel linear RNN stack. The proposed approach integrates a learnable spectral decomposition of the input signal with a multilayer perceptron (MLP) as measurement functions. SKOLR is evaluated on standard forecasting benchmarks and nonlinear dynamical systems, where it appears to achieve competitive results.

**Claims And Evidence:**

The paper makes several key claims:
1. Establishing a Connection Between Koopman Operator Theory and Linear RNNs – The authors show an equivalence between Koopman operator approximations and linear RNN updates, but this connection is somewhat superficial. First, this is not a strictly novel result, as a very similar connection was shown in [a] (see appendix E). Furthermore, I think that theoretical justification is not fully developed, and the assumptions required for this equivalence to hold are not explicitly stated.

2. Superior Forecasting Performance – While the reported numbers show competitive results, the claim of superiority cannot be convincingly ascertained. Many of the performance gains are on the third significant digit, and since variances are not reported, the differences between methods might be due to statistical uncertainty.

3. Efficiency and Parameter Reduction – The authors show that SKOLR achieves competitive results with reduced computational costs.

Overall, the claims lack sufficient theoretical and empirical justification, and the evidence presented does not unambiguously support SKOLR’s superiority.

[a] "Resurrecting Recurrent Neural Networks for Long Sequences" by Orvieto et al. 2023

**Essential References Not Discussed:**

For linear RNNs, please check the aforementioned [a], establishing the connection between Linear RNNs and koopman operators. More generally, there is a growing literature on machine learning methods for Koopman operators, see e.g. [b]


[a] "Resurrecting Recurrent Neural Networks for Long Sequences" by Orvieto et al. 2023
[b] "Learning dynamical systems via Koopman operator regression in reproducing kernel Hilbert spaces" by Kostic et al. 2022

**Experimental Designs Or Analyses:**

I think that the analysis of the method is far from comprehensive. Indeed, SKOLR is an assembly of many inter-dependent components:
1. Spectral encoder and spectral decoder
2. Structured RNN stack
While an "Ablation Study" is presented in Section 4.3 the Authors mainly test how different settings in the encoder and RNN stacks impact the final performance. I would classify it as a study of the dependence on hyperparameters. An ablation study, conversely, should selectively remove components (such as the spectral encoder, or the structured decomposition of the RNN) and see the impact on performances.

What Section 4.3 shows, instead, is that if the hyperparameters of SKOLR are not well-tuned, the claimed performance gains are completely washed away.

**Methods And Evaluation Criteria:**

The evaluation criteria largely follows [b], and tests SKOLR on standard time-series forecasting benchmarks. As I already mentioned, a big shortcoming of the evaluation is the lack of reported standard deviations. As many of the baselines attain similar errors, single estimates of the mean absolute error are, in my opinion, not sufficient to evaluate a method.

[b] "Koopa: Learning non- stationary time series dynamics with koopman predictors." by Liu et al. 2023

**Other Comments Or Suggestions:**

- More rigorous ablations should be included to demonstrate the necessity of each architectural component.
- The paper should engage more deeply with prior literature to clarify its novelty.
- More discussion on failure cases and limitations would improve transparency.

**Other Strengths And Weaknesses:**

Strengths:
- The paper provides a method for time-series forecasting connecting Koopman theory with RNNs.
- The empirical results show competitive performance, even if not definitively superior.
- The method is computationally efficient compared to transformers.

Weakneses
- The theoretical claims are underdeveloped and were already presented in prior works, which the Authors do not acknowledge.
- The experimental results do not convincingly demonstrate clear superiority over existing methods.

**Questions For Authors:**

N/A

**Relation To Broader Scientific Literature:**

This paper does not sufficiently engage with prior work. Koopman-based learning is an active research area, yet the paper misses key references on data-driven Koopman approximations and alternative learning-based Koopman methods. See the essential references not discussed, below.

**Theoretical Claims:**

As mentioned above, the main theoretical claim of the paper is the equivalence between linear RNNs and Koopman operator theory. For this equivalence, only intuition is provided, there is no formal theorem or clear set of assumptions under which the approximation holds. Furthermore, a very similar equivalence was proved already in [a].

[a] "Resurrecting Recurrent Neural Networks for Long Sequences" by Orvieto et al. 2023

---

> ### Author Rebuttal · Authors · 2025-04-01
>
> ## Claim 1
> We would kindly ask the reviewer for more supporting arguments for the assertions.
>
> > "connection is somewhat superficial"
>
> We establish a clear connection between an EDMD-style approximation of the Koopman operator for a history-augmented state and a linear RNN. This motivates a highly effective but simple architecture, which turns out to be not superficial.
>
> > "theoretical justification is not fully developed"
>
> We contend that we have fully developed the theoretical justification mathematically.
> Eq. 5 is the equation for a linear RNN with a finite history. Eq. 6-7 present the EDMD-style approximation to the Koopman operator. And Sec. 3.3 explains that if we consider an augmented state with historical observations then we can rewrite the approximate Koopman operator relation as Eq. 8. Obviously, Eq. 8 has the identical structural form as Eq. 5, implying that we can implement Eq. 8 using a linear RNN (with an MLP embedding).
>
> > "only intuition is provided"
>
> The development is by no means "only intuition". It is a clear and concise mathematical development.
> However, we are happy to provide a more formal proposition-proof structure in an appendix if this is desired.
> Please flag this, and we will provide it in the final response.
>
> > "assumptions ... are not explicitly stated"
>
> We kindly ask which assumptions are missing in this development? The key assumptions are expressed in the fact that we are applying EDMD-style approximation (so the EDMD assumptions apply); we specify the full rank assumption for a unique solution.
>
> > "a very similar equivalence was proved already"
>
> App. E [a] contains NO equations connecting linear RNNs to Koopman theory — only a **qualitative observation ("Sounds familiar? This is exactly what a wide MLP + Linear RNN can do.")** without mathematical formulation or proof. While we acknowledge this prior insight, our contribution provides the explicit mathematical connection that was not previously established.
>
> ## Connection to [a] and [b]
> We would include the suggested works into our discussion.
>
> However, we'd like to highlight key distinctions from [a]. As aforementioned, [a] only offers a qualitative discussion of Koopman operator theory relevance, listing equations for Koopman theory (Appx. E, Eq.s 17-22) followed by a conclusion without any mathematical development.
> More importantly, the architecture in [a] does not follow the qualitative observation in Appex. E: the stack of [linear recurrent units + MLPs] has strayed away from the Koopman operators.
> In contrast, our architecture is an exact match to our derivations in (5) and (8).
>
> Furthermore, [a] does not address time-series forecasting with any experiments on this task. In fact, the proposed architecture in [a] performs poorly in the time-series forecasting setting (See [Tab. 7](https://anonymous.4open.science/r/SKOLR-1D6F/Tab7_LRU.pdf)).
>
> Concerning [b], we will cite it and include more discussion about the intersection of machine learning methods and Koopman operators.
>
> ## Statistical Significance
> To address the reviewer's concern about statistical significance, we add [Tab. 6](https://anonymous.4open.science/r/SKOLR-1D6F/Tab6_Std.pdf) with standard deviation (std) across 3 independent runs for all datasets and forecast horizons. The low stds (<0.003 for most datasets) demonstrate the consistency and reliability of SKOLR's performance.
> Thus, we contend that SKOLR's improvements, even when small in magnitude, represent genuine performance differences rather than statistical fluctuations.
> Additionally, we will provide confidence intervals and statistical significance tests in our final response.
>
> ## Efficiency and Parameter Reduction
> We will add a dedicated "Theoretical Complexity Analysis" section to formally establish SKOLR's computational benefits. Due to the word limit, please refer to response to reviewer hUoy (point 1.2) for detailed information.
> This comprehensive theoretical analysis, combined with our expanded empirical results in [Tab.4](https://anonymous.4open.science/r/SKOLR-1D6F/Tab4_Computation.pdf), provides thorough justification for SKOLR's efficiency claims.
>
> ## Ablation study
> As suggested by the reviewer, we have also conducted a more comprehensive ablation study on the design elements of SKOLR.
> As shown in [Tab. 5](https://anonymous.4open.science/r/SKOLR-1D6F/Tab5_Ablation.pdf), we compare our full SKOLR model with two ablated variants: (1) "w/o Structure": no structured decomposition, using a single branch with dimension (N×D); (2) "w/o Spectral Encoder": no learnable frequency decomposition, while maintaining the multi-branch structure.
>
> The results show that both components contribute meaningfully. Removing the structured decomposition leads to performance degradation on 27/32 tasks, with notable declines on ETTh1 and ILI, while increasing computational overhead. Similarly, removing spectral encoder impacts performance on 23/32 tasks, though with a smaller overall effect.

---

### Official Review · Reviewer_fgpm · 2025-03-11

**Overall Recommendation:** 3

**Summary:**

The authors develop an approach to forecast time-series, via Koopman operator theory, through the use of linear RNNs. The authors demonstrate that their approach delivers state-of-the-art performance on long and short term forecasting benchmarks and is significantly less computationally expensive.

**Claims And Evidence:**

There are a few claims that I believe could be strengthened with more convincing evidence/change of language:

1. That the different branches of SKOLR "specialize in distinct frequency bands" (page 6). Looking at Fig. 3 (left column), it is not particularly clear to me that the two branches "specialize in distinct frequencies". If anything, it looks like they agree on many of the frequencies. Clarifying this (and/or providing a quantitative comparison to make their point more clear) would be good.

2. The performance of SKOLR on ILI is the best of all the methods tested. However, saying "SKOLR demonstrates remarkable forecasting capability", when the MSE (~1.5) is of a similar magnitude as the average ILI across a given flu season (e.g., the national baseline ILI for the 2018-2019 season was 2.2), is perhaps a little misleading. Toning down this remark is, in my opinion, more appropriate.

3. In the ablation study (setting the learnable weights to 1) the results appear relatively consistent with the non-ablated model (the difference between the MSE is often different by <5%). This combined with the first point above (#1) makes me somewhat skeptical that weighting the branches differently really affects the performance significantly. Doing the same analysis on other data sets where perhaps it makes a bigger difference or changing the language to be more transparent that the weighting doesn't often lead to a large difference would be good.

**Essential References Not Discussed:**

Given that this area of using Koopman + ML for forecasting has exploded over the past few years, it is reasonable that there may be papers that the authors did not cite that I believe could be relevant. However, there are several foundational papers that the authors did not cite. Adding these (and discussing how the author's work relates) is important and will strengthen the paper.

1. Mezic 2005 Nonlinear Dynamics - This work is the most relevant paper for discussing Koopman mode decomposition and the Koopman representation (and should be cited when citing Koopman 1931).
2. Rowley et al. 2009 Journal of Fluid Mechanics - This was the first work to demonstrate that DMD could approximate the Koopman mode decomposition.
3. Kutz et al. 2015 SIAM J. Appl. Dyn. Sys. - This work proposes a similar decomposition of data via FFT and I believe it employs a similar block structure as is used in SKOLR.
4. Arbabi and Mezic 2017 SIAM J. Appl. Dyn. Sys. - This work was the first to rigorously propose using time-delays for constructing a Koopman operator (Hankel DMD).

Also, minor point, but the authors reference Li et al. (2017) when talking about EDMD (Sec. 3.2), but the first paper to develop EDMD was Williams et al. (2015) (which the authors do cite earlier and later).

**Experimental Designs Or Analyses:**

The experimental design and analysis all appears sound.

**Methods And Evaluation Criteria:**

The authors did a great job evaluating their approach. They thoroughly compared on baselines and provided evaluation of model size and computational resources used.

**Other Comments Or Suggestions:**

**Minor points**
1. (very minor) The authors write "time series" throughout the paper, but the title has "time-series"
2. (very minor) The authors write that "early methods used RNNs and CNNs" (page 8), but then cite papers from 2020/2024.

**Other Strengths And Weaknesses:**

**Strengths**
1. This paper achieves SOTA performance (even if the gains are at times relatively small). The comparison with other methods was convincing.
2. The improvement in decreasing model size was significant and is an exciting outcome of the work.
3. Fig. 1 was helpful for understanding the method.
4. Comparing how the use of different number of branches affected performance (Fig. 4, Tables 3 and 4) was interesting.

**Weaknesses**
1. As noted above, there are several claims that I think need to be clarified/strengthened.
2. As noted above, there are several important papers that need to be cited and discussed.
3. As noted above, clarity is need when discussing the "equivalence" with the linear RNNs.

**Questions For Authors:**

I do not have any other questions for the authors.

**Relation To Broader Scientific Literature:**

The contributions of this paper are relatively well situated within the broader scientific literature. In particular, by comparing their method to Koopa, the current SOTA, they convincingly show some of the benefits of their approach. Discussing, in more detail, the advantage of having a smaller, less computationally demanding model (in the Introduction) could further strengthen their paper.

**Theoretical Claims:**

The main theoretical claim made by this paper (which is perhaps too strong of a word, as the authors more argue that two things are similar) is that the Koopman representation can be equivalent to a linear RNN. This inspires their branched decomposition. While I agree that Eq. 5 and Eq. 8 are the same, I think this claim is somewhat misleading. In particular, RNNs typically take in an input at each time step (the $U v_k$ in Eq. 4), in addition to propagating forward the internal state. This makes linear RNNs a dynamical system with control (or inputs). In Eq. 8, the Koopman operator is solely propagating forward the observable functions $g$. This is more equivalent to inputing some state into a linear RNN and then evolving it forward (without providing any more inputs). So while I think the equivalence is correct, it relies on viewing RNNs differently than they are typically thought of. Making this more clear is important.

---

> ### Author Rebuttal · Authors · 2025-04-01
>
> ## Strengthen Claims with more Convincing Evidence
>
> ### Branch Decomposition - frequency band specialization
> We appreciate the reviewer’s insightful feedback and agree that "specialization" is too strong a claim. We consider that Fig. 3 provides evidence that different branches place more emphasis on some frequency bands (e.g., Branch 1 has a greater relative emphasis on the low-frequency component). But there is not evidence of specialization.
>
> ### ILI performance
> Regarding the performance of SKOLR on ILI, we recognize the concern about the phrasing of "remarkable forecasting capability."
> While SKOLR achieves the best MSE among the tested models, we agree that this is mischaracterization, given that the error is still high. We will change the text to more accurately state that SKOLR outperforms the baseline methods, with significant error reduction for the shorter horizons.
>
>
> ### Ablation Study - learnable frequencies
> The observation on the ablation study is insightful. We agree with the general observation that the learnable decomposition is not a critical component and does not lead to a major improvement. On the other hand, it does lead to a relatively consistent 2-5\% improvement, and it is not many parameters to learn. We will modify the text to downplay the significance of the learnable frequency decomposition, and explain that while it does bring some benefit, SKOLR performs well without it. It suggests that the key aspect of the multiple branches is to allow for MLPs to learn different embeddings.
>
> ## Theoretical Claims - linkage between Koopman operator and RNN
>
> Our claim is not as strong as "the Koopman representation can be equivalent to a linear RNN". In the abstract and introduction, we were careful to write that we "establish a connection between Koopman operator approximation and linear RNNs". In Section 3.3, after equation 8, we write that "this structure can be implemented as a linear RNN". **So, our conclusion/claim is more that a linear RNN (with MLP embeddings) provides an efficient and effective mechanism for implementing a specific structured Koopman operator approximation (involving an augmented state incorporating past observations).** Our equivalence specifically refers to the scenario where we construct an extended state representation using lagged observations (Section 3.3), which effectively transforms the Koopman operator approximation into a form where Eq. 5 and Eq. 8 become structurally identical.
>
> Since your review is thorough, indicating careful attention to the paper, it seems that we haven't made it clear enough that our theoretical claim (which is indeed more of an algorithmic development) is more limited. We will modify the language to make it very clear. This is perhaps also an issue we would raise with the qualitative connection in Orvieto et al. (2023) between Koopman operator analysis and an MLP + linear RNN; when you write equations, the connection is not quite so simple, nor as general, as suggested in that work.
>
>
>
> ## References
> Thank you very much for highlighting these essential references. We will incorporate them to properly situate SKOLR within the Koopman literature. We'll add Mezic (2005) alongside Koopman (1931) as foundational work on Koopman mode decomposition, and Rowley et al. (2009) when discussing DMD's relationship to Koopman theory. We'll discuss connections between our spectral decomposition approach and Kutz et al. (2015), whose frequency-based decomposition shares similarities with our branch structure. Importantly, we'll acknowledge Arbabi and Mezic (2017) when discussing time-delay embeddings in Section 3.3, as their Hankel DMD provides theoretical support for our extended state construction. We'll also correctly attribute EDMD to Williams et al. (2015) in Section 3.2, while maintaining the Li et al. (2017) reference for neural network extensions. These additions will provide proper attribution and better contextualize our contributions within the evolving landscape of Koopman-based methods.
>
>
> ## Weaknesses
>
> The three weaknesses are addressed in the comments above.
>
> ## Suggestions
>
> Thank you for these careful observations. We will standardize our usage of "time series" (without hyphen) throughout the paper for consistency.
>
> We also appreciate you flagging the timeline in our literature review. We will revise this section to include pioneering works such as [1]-[3] and properly reflect the development of time series forecasting methods.
>
> [1] Hochreiter, S., \& Schmidhuber, J. (1997). Long short-term memory. Neural Computation, 9(8), 1735-1780.
> [2] Chung, J., Gulcehre, C., Cho, K., & Bengio, Y. (2014). Empirical evaluation of gated recurrent neural networks on sequence modeling.
> [3] Binkowski, M., Marti, G., & Donnat, P. (2018, July). Autoregressive convolutional neural networks for asynchronous time series.

---

> > ### Comment · Reviewer_fgpm · 2025-04-07
> >
> > I thank the authors for their thorough and respectful responses to my review. It is clear they are willing to engage in the review process. All of my comments/questions have been sufficiently addressed and I believe the paper, with the changes the authors detail that they will make, will strengthen the paper. At the moment I will keep my score of a 3, but I feel more confident in this assessment and more confident in being willing to support this paper.

---

> > > ### Author Response · Authors · 2025-04-09
> > >
> > > Thank you for your thoughtful feedback and for acknowledging our engagement with the review process. We're pleased that our responses have addressed your comments and questions satisfactorily.
> > >
> > > We appreciate your continued support of our paper. The revisions we've outlined will be implemented carefully to strengthen the paper as you've noted. Your constructive criticism has been invaluable in helping us improve our manuscript, and we're grateful for the time and expertise you've contributed to this process.

---

### Official Review · Reviewer_hUoy · 2025-03-13

**Overall Recommendation:** 3

**Summary:**

This paper introduces a novel approach that connects Koopman operator approximations with RNNs for efficient time-series modeling. Koopman operator theory provides a linear representation of nonlinear dynamical systems but is typically infinite-dimensional, making it challenging to apply directly. To address this, the authors propose a structured approximation of the Koopman operator and establish its equivalence to linear RNN updates. Based on this insight, they develop SKOLR, a model that integrates learnable spectral decomposition of input signals with multilayer perceptrons (MLPs) as measurement functions and a highly parallelized linear RNN stack to implement the structured Koopman operator.

## update after rebuttal
Sorry for the last-minute response. The authors have addressed most of my concerns. I have updated my score to "weak accept" accordingly.

**Claims And Evidence:**

The claims in the paper are generally well-supported by clear and convincing evidence. However, there are still some problematic claims as follows:

* High efficiency: the authors claim in there paper that the proposed architecture is computationally effective. However, they only provide experiment results in ETTm2 and Traffic datasets (in Fig. 5). The authors should include results of other datasets. Moreover, including theoretical analysis of computation complexity could strengthen this claim. The authors also mention that the branch decomposition make the linear RNN chains highly parallel. I suggest that the authors should also explain how they implement the parallel computing.
* Exceptional performance: the authors claim that this architure delivers exceptional performance. However, I do not showcases in the paper. The authors should also present showcases to demenstrate that their model are effective in capturing complex temporal patterns (see line 366).

**Essential References Not Discussed:**

There is some reseearches incorporating Koopman Theory into RNNs and the authors should briefly discuss the difference and novelty.

[1] Recurrent neural networks and Koopman-based frameworks for temporal predictions in a low-order model of turbulence

**Experimental Designs Or Analyses:**

I have checked the soundness/validity of experimental designs and analyses. The authors provide results in widely used time series forecasting benchmarks and nonlinear dynamical system and claim performance of the model and provide efficiency comparision in ETTm2 and Traffic to claim the efficiency. As discussed above, more showcases should be included and efficiency promotion should be analyzed in more datasets.

**Methods And Evaluation Criteria:**

The proposed methods and evaluation criteria in the paper are well-suited for the problem. The authors bridge a gap between Koopa Operator Theory and RNN to achieve both efficiency and performance. The evaluation criteria (ECL, Traffic and other benchmarks & metrics) is widely used in time series forecasting evaluation.

**Other Comments Or Suggestions:**

I am willing to raise my score if the authors could address my question.

**Other Strengths And Weaknesses:**

Strengths:

* Writting: This paper is well written and the figures could express authors' idea clearly. After reading this paper, I could easily understand the authors' idea. The work is well written providing sufficient relevant background knowledge on Koopman operator theory also for reader who is unfamiliar with this theory.
* Strong performance in benchmarks: SKOLR outperforms or matches state-of-the-art models on multiple standard forecasting datasets. The model also demonstrates superior performance in nonlinear dynamical system modeling (e.g., Pendulum, Duffing, Lotka-Volterra, Lorenz ’63), validating its effectiveness beyond traditional forecasting tasks.

Weaknesses:

* Clarity: While the learnable frequency decomposition improves performance, it is not clearly justified why this approach is superior to other feature extraction methods. The paper lacks an in-depth exploration of why the Koopman-based linear RNN structure generalizes well to different datasets beyond empirical results.
* Theoretical analysis: the essence of the deep Koopman method lies in the spectrum of the Koopman operator, because the eigenvalues determine the model's behavior during long-term evolution. However, the author did not provide any analysis or visualized results regarding the eigenvalues.

**Questions For Authors:**

1. I noticed in Fig. 1, the Encoder only untilizes $x_1$ to derive $z_i$ for $i=1,2...L$, is it a typo?
2. Will this architecture suffer from error accumulation which usually happens in RNNs? How to solve this problem? Could authors provide some showcases for long context prediction?

**Relation To Broader Scientific Literature:**

The key contributions of this paper are related to the application of physics informed methods (Koopman Operator Theory in this paper) in time series. The proposed decomposition and RNN stack improve computational efficiency and accuracy compared to previous Koopman-based methods ([1] [2]).

[1] Koopa: Learning Non-stationary Time Series Dynamics with Koopman Predictors.

[2] Koopman neural forecaster for time series with temporal distribution shifts.

**Theoretical Claims:**

I have checked the correctness of proofs for theoretical claims. If authors could also include theoretical analysis for computational complexity (discussed above), it would be better.

---

> ### Author Rebuttal · Authors · 2025-04-01
>
> ## 1. High Efficiency
> ### 1.1 Computational efficiency:
> We provide additional results for all datasets. Please see [Tab.4](https://anonymous.4open.science/r/SKOLR-1D6F/Tab4_Computation.pdf). SKOLR achieves a compelling trade-off between memory, computation time, and accuracy.
>
> ### 1.2 Theoretical Complexity Analysis
> SKOLR achieves computational efficiency through structured design and linear operations. For a time series (length L, patch length P, embedding dimension D, N branches):
>
> - **Time complexity**: O(N × (L/P) × D²) from spectral decomposition, encoder/decoder MLPs, and linear RNN computation
> - **Memory complexity**: O(N × D²) for parameters and O(N × (L/P) × D) for activations
>
> Compared to a non-structured model with dimension D' = N×D:
> - Non-structured approach: O((L/P) × N²D²) time and O(N²D²) memory
> - SKOLR provides N-fold reduction in computational requirements
>
> SKOLR avoids quadratic scaling with sequence length seen in transformers (O((L/P)² × D + (L/P) × D²) time, O((L/P)² + (L/P) × D) memory).
>
> ### 1.3 Parallel computing
> The N separate branches are processed independently (in our code), reducing time complexity to O((L/P) × D²).
>
> The linear RNN computation has no activation functions, so the hidden state evolution is: $h_k = g(y_k) + \sum_{s=1}^{L/P} W^s g(y_{k-s})$. This allows efficient matrix operations, reducing time complexity to O($D^3 \log(L/P)$ + $(L/P)^2 \times D$) per branch. For time series where $L/P \ll D$, this is a significant speedup.
>
> ## 2. Exceptional Performance Claim
> Our claims are based on Tab.s 1 \& 2 and Fig.s 2 \& 4. SKOLR requires less-than-half the memory of any baseline and less-than-half of Koopa's training time (Fig.4). SKOLR has the (equal-)lowest MSE in 17 our of 32 tasks and ranks second in a further 7 (Tab.1). SKOLR significantly outperforms Koopa for synthetic non-linear systems (Tab.2). Given the very low memory footprint, the low training time, and the impressive accuracy, we consider that the claim of "exceptional" performance is supported, but we can use a less strong adjective.
>
> We agree that the paper would be strengthened by more examples (showcases) demonstrating the capture of complex patterns. We do have one example in Fig. 3, but we will include more. Please see [Fig. 2](https://anonymous.4open.science/r/SKOLR-1D6F/Fig2_Compare.pdf) as an example. SKOLR's predictions have much lower variance than Koopa's and track the oscillatory behaviour better.
>
> ## 3. Relevant Reference
> We will cite it and add discussion. The paper evaluates the existing non-linear RNNs and Koopman methods for near-wall turbulence. It does not draw connections or develop a new method.
>
> ## 4. Clarity
> ### 4.1 Frequency Decomposition
> We agree that the motivation for our learnable frequency decomposition could be better. Please see the response to Reviewer kXAS. Learnable frequency decomposition offers three key advantages. (1) Each branch can focus on specific frequency bands, decomposing complex dynamics. (2) Learnable components adaptively determine which frequencies are most informative for prediction. (3) This approach aligns with Koopman theory, as different frequency components often correspond to different Koopman modes.
>
> ### 4.2 Generalization capability
> Constructing theoretical guarantees that the approach generalizes is challenging and would be a paper all on its own. We do provide experimental results for a large variety of time-series that exhibit very different characteristics, ranging from strongly seasonal temperature series to pendulums exhibiting highly non-linear dynamics. We consider that the experimental evidence in the paper is strongly supportive of a capability to generalize to a diverse range of time series.
>
> ## 5. Theoretical analysis: eigenvalues
> This is an excellent suggestion. Our focus is forecasting, so our results and analysis concentrate on that task. However, the analysis of learned Koopman operator eigenvalues can indeed reveal important characteristics.
>
> We analyzed eigenvalue plots for Traffic dataset (see [Fig.3](https://anonymous.4open.science/r/SKOLR-1D6F/Fig3_Eigenvalue.pdf)). We see that each branch learns complementary spectral properties, with all eigenvalues within the unit circle, indicating stable dynamics. Branch 1 shows concentration at magnitude 0.4, while Branch 2 exhibits a more uniform distribution. The presence of larger magnitudes (0.7-0.9) indicates capture of longer-term patterns.
>
> ## 6. Questions
> ### 6.1
> It is a typo. We will revise Fig. 1 to show the proper time indexing across all components.
>
> ### 6.2 Long horizon prediction
> To show SKOLR's capability for longer horizons, we included experiments with extended prediction horizons (T=336, 720) in App. B.2 (Tab. 9), SKOLR maintains its performance advantage at extended horizons, with lower error growth rates.
>
> Please also see response to reviewer kXAS concerning test-time horizon extension and [Tab. 1](https://anonymous.4open.science/r/SKOLR-1D6F/Tab1_ScaleUp.pdf).

---

> > ### Comment · Reviewer_hUoy · 2025-04-03
> >
> > Thanks for your response. My questions are still not completely solved. I will appreciate it if authors could provide more explanations.
> > 1. For the showcases, I think authors should evaluate proposed model in real-world non-stationary datasets, rather than those that could be simulated using some functions.
> > 2. For theoretical analysis of eigenvalues, could authors explain what the results mean?
> > 3. My question "Will this architecture suffer from error accumulation which usually happens in RNNs? How to solve this problem?" is still not answered.

---

> > > ### Author Response · Authors · 2025-04-09
> > >
> > > Dear Reviewer hUoy,
> > >
> > > We appreciate your thoughtful questions. All figures in this response can be found in this [link](https://anonymous.4open.science/r/SKOLR-1D6F/Fig4-9.pdf).
> > >
> > > ## Showcase
> > > We add examples for ETTm2 and Electricity to showcase SKOLR's branchwise behavior. The figures show distinct frequency specializations. In Fig. 4, SKOLR decomposes the time series into complementary frequency components, with Branch 1 focusing on lower frequencies while Branch 2 captures specific dominant frequencies.
> > >
> > > In Fig. 5, Branch 1 emphasizes lower frequency components, with pronounced content around 2 μHz representing weekly cycles and 12-13 μHz capturing daily patterns. In contrast, branch 2 gives greater emphasis to higher frequency components, particularly around 37 μHz, which corresponds to shorter intraday cycles occurring every 6-8 hours. The time-domain plots demonstrate how these spectral differences translate into signal reconstruction. Branch 1 includes more longer-term cycles (daily/weekly) while Branch 2 has more emphasis on intra-day structure.
> > >
> > > ## Eigenvalues
> > > Thank you for your question about the theoretical implications of our eigenvalue analysis.
> > >
> > > The Koopman modes are derived through eigendecomposition of the RNN weight matrices $M_i$. These modes represent fundamental dynamical patterns in the data. Each mode captures specific components of the time series. The stability and oscillatory behavior of each mode is determined by the corresponding eigenvalue’s position in the complex plane.
> > > The eigenvalue plots in Fig.6 (last rebuttal) show that our Koopman operator approximation maintains stability by ensuring all eigenvalues remain within the unit circle.
> > >
> > > To expand the analysis, we include a case study by constructing the prediction from a single dominant Koopman mode. As shown in Fig. 7, the dominant mode successfully captures the primary trend and approximate shape of the future values, though with some timing and amplitude differences. While the dominant mode identifies the primary oscillatory behavior, the complete dynamics require contributions from multiple Koopman modes.
> > >
> > > This observation also partially addresses the next question. Error divergence is characteristic of unstable systems. Since our learned system maintains stability with eigenvalues inside the unit circle, error effects tend to be dampened over time.
> > >
> > > ## Error Accumulation
> > > Thank you for raising this important question about error accumulation. Our SKOLR model addresses this common RNN limitation through multiple complementary approaches:
> > >
> > > **1. Patch-based Processing**
> > >
> > > Time series forecasting has two main prediction methods:
> > > (a) direct prediction: forecasts the entire horizon at once but is parameter-inefficient and cannot extend the prediction horizon after training;
> > > (b) recursive prediction: iteratively uses predictions as inputs but may suffer from error accumulation over long sequences.
> > >
> > > In SKOLR, we use a patching approach (App.A.2) to create an effective middle ground between these methods.
> > > Instead of operating at the individual timestep level, we work with patches of multiple timesteps, directly predicting all values within each patch while only applying recursion between patches. This dramatically reduces the number of recursive steps (e.g., from 720 to just 5 with patch length 144), controlling error accumulation. Additionally, this method also reduces complexity to $O(\frac{L}{P})$) from RNN timestamp-based approaches ($O(L)$) while maintaining the core principle of Koopman operator theory.
> > >
> > > **2. Experiments on long horizon with varing iteration steps**
> > >
> > > We have conducted an experiment for longer horizon $L=720, T=720$ on dataset ETTm2. We vary the patch length $P=\{16, 24, 48, 144, 240\}$ of SKOLR to see the effect of the number of recursive steps.
> > >
> > > SKOLR with P=16 requires 45 recursive steps, while P=240 needs only 3, yet they maintain comparable error profiles in Fig.8. This empirically demonstrates that SKOLR's patch-based approach effectively controls error accumulation, even with increased recursion.
> > >
> > > Our experiments for T={336,720} (Appx. B.2 Table.9) has P=16. Even with an increased number of recursions, SKOLR emerges as the leading performer.
> > >
> > > **3. Comparison with Non-recurrent Models**
> > >
> > > We also compared SKOLR with iTransformer, which performs direct prediction without recurrence. Both models show similar patterns of error increase with longer horizons, indicating that this modest increase is inherent to all forecasting approaches when extending the prediction range, rather than being caused by recurrent error accumulation.
> > >
> > > **4. Visual Evidence**
> > >
> > > Our visualizations on horizon $T=720$ do not show obvious differences between the first and last patch predictions, demonstrating that error growth remains well-controlled even across extended forecasting horizons. See Fig.9 as an example.
> > >
> > > **These empirical results confirm that our architecture successfully overcomes the error accumulation limitations.**

---

### Official Review · Reviewer_kXAS · 2025-03-16

**Overall Recommendation:** 3

**Summary:**

This paper proposes a new linear RNN for time-series forecasting inspired by Koopman operator theory. The problem setup is, for an autonomous dynamical system $\mathbf{x}\_{k+1}=F(\mathbf{x}\_k)$ with observable $\mathbf{y}\_k=h(\mathbf{x}\_k)$ for an unknown $h(\cdot)$, to condition on a partial trajectory $\mathbf{y}\_1,...,\mathbf{y}\_L$ of a fixed length $L$ and forecast $T$ steps of the future $\mathbf{y}\_{L+1},...,\mathbf{y}\_{L+T}$. The authors start with the standard form of a linear RNN which evolves its hidden state $\mathbf{h}_k$ conditioned on an input sequence $\mathbf{v}_k$, and consider the case where $\mathbf{v}_k$ is pointwise mapped from $\mathbf{y}_k$. This reduces the linear RNN to $\mathbf{h}\_k=\sum\_{s=1}^k\mathbf{W}^{k-s} g(\mathbf{y}\_s)$ for some $g(\cdot)$. Then, the authors propose to construct an approximate state as a stack of past $L$ measurements  $\tilde{\mathbf{x}}\_k \coloneqq [\mathbf{y}\_{k-L+1},...,\mathbf{y}\_k]$ and learn the dynamics $\tilde{\mathbf{x}}\_k=\tilde{F}(\tilde{\mathbf{x}}\_{k-1})$ by jointly learning $g(\cdot)$ on $\tilde{\mathbf{x}}\_k$ and a matrix $\mathbf{M}$ which parameterizes the dynamics of $g(\tilde{\mathbf{x}}\_k)$, using a similar objective to the standard EDMD, but with a structured constraint on $\mathbf{M}$ such that the learned dynamics can be always written as $g(\tilde{\mathbf{x}}\_k) = \mathbf{h}\_k=\sum\_{s=1}^k\mathbf{W}^{k-s} g(\mathbf{y}\_s)$. Basically, this constraint means that $\mathbf{M}$ is a blockwise diagonal matrix with each block given as a power of some learnable matrix $\mathbf{W}$. The authors parameterize $g(\cdot)$ as trainable gating in frequency domain (FFT -> trainable gate per frequency -> IFFT) followed by pointwise MLP, at multiple "heads". Then at each head the structured matrix $\mathbf{M}$ is separately parameterized. The decoder is simply a positionwise MLP. The authors report state-of-the-art performance of the proposed architecture on 8 benchmark datasets, using $L=2T$, and 4 physical dynamical systems. Further analysis and ablation study on the role of each component and hyperparameter are provided.

## update after rebuttal

The following is my revised understanding of the paper:

- The paper proposes a new linear RNN for time-series forecasting. The key components proposed are block-diagonal state-transition matrix and learnable frequency-domain gating. The former is inspired by Koopman operator theory for deterministic nonlinear dynamical systems and the EDMD algorithm. The latter is inspired by existing works in time-series modeling leveraging frequency-domain analysis components.

- Empirically, the proposed method overall shows a competitive performance compared to the state-of-the-art Koopa, and at the same time has a good time and space computational efficiency.

- A weakness is that while Koopman operator theory is stated as the inspiration of the architecture, the frequency-domain gating is not motivated from its principles. Therefore it is not clear what the final model is implementing in the perspective of Koopman theory.

Considering both the practical benefits and weaknesses in the theoretical analysis, I updated my score accordingly.

**Claims And Evidence:**

The paper is focused on demonstrating state-of-the-art performance of the proposed method, and the main results are given in Table 1 and Figure 2, which supports the main claim. Also, the claim in page 5 that multi-branch parameterization reduces the parameter count is supported by Section 4.3.1. On the other hand, limitations of the presented evidences are as follows:

- The authors have not shown whether it is possible to use a longer forecast horizon at the test-time, as in Section 5.3 of Liu et al., (2023). The utility of the method would be more convincing if this capability is demonstrated, especially given that both Koopa and SKOLR are based on Koopman operator theory.
- In Appendix A.2, the authors claim they adopt non-overlapping patch tokenization after trainable gating in frequency domain. It is not clear how much this impacts the model efficiency compared to not tokenizing. Also, it is unclear whether this can be applied to other baselines (e.g., Koopa) as well. For fair comparison of efficiency one would expect patch tokenization to be consistently applied (or not applied) to all methods.
- The hyperparameter selection protocol in Appendix A.2 is not very clearly described. Is the validation split used for the grid search? Also, having some analysis on the sensitivity to the hyperparameter choice (e.g., Appendix C of Liu et al., (2023)) would make the results more convincing.
- In Table 8, some of the numbers are also not consistent with the reported results in Liu et al., (2023), and the highlightings of the MASE metric for Quarter and Others are wrong.

Liu et al. Koopa: Learning Non-stationary Time Series Dynamics with Koopman Predictors (2023)

**Essential References Not Discussed:**

The observation that Koopman operator theory can be connected to linear RNNs is not new, for example we have Orvieto et al. Resurrecting Recurrent Neural Networks for Long Sequences (2023), specifically Appendix E.1.2.

**Experimental Designs Or Analyses:**

Please see the above sections.

**Methods And Evaluation Criteria:**

The evaluation criteria closely follows a prior work (Liu et al., (2023)) which seems sound to me.

**Other Comments Or Suggestions:**

For the presentation in Section 3, I suggest that the authors first describe the whole pipeline (including the encoder and decoder) for a single branch ($N=1$), and then describe the multiple-branch case. This is what is often done in sequence modeling papers using multi-head architectures.

**Other Strengths And Weaknesses:**

Strengths
- The empirical results are promising, especially considering the computational efficiency of the proposed method (but see the concerns in the Claims And Evidence section).

Weaknesses
- The presentation of the methodology in Section 3 could be overall improved. Especially, while the authors have invested a significant amount of text describing Koopman operator theory, linear RNNs, and EDMD, the design of the encoder which uses trainable frequency-domain gating is not motivated concretely from the background theory and seems rather arbitrary. This is a key design choice with its own ablation study, but I do not see how this is motivated. Also, the mix-up of symbols (e.g., using $\mathbf{Y}$ both before and after the frequency-domain gating) makes the description quite confusing.

**Questions For Authors:**

- In Section 4.2, are there particular reasons to refer to these 4 systems as non-linear systems, given that the dynamics in Table 1 are also non-linear?
- In Figure 3, what parts of the figure do (a) and (b) correspond to?

**Relation To Broader Scientific Literature:**

The paper is related to the fields of linear RNNs, time-series forecasting, and Koopman operator theory. Specifically, the contributions are in the application of Koopman operator theory for designing a linear RNN for time-series forecasting.

**Theoretical Claims:**

The paper does not make theoretical claims.

---

> ### Author Rebuttal · Authors · 2025-04-01
>
> ## Longer test-time forecast horizon
> We have now conducted experiments for increased test-time horizon. Please see [Tab. 1](https://anonymous.4open.science/r/SKOLR-1D6F/Tab1_ScaleUp.pdf). SKOLR has a recursive structure. Even if we train over a given horizon, we can recursively predict for a longer horizon. There is performance deterioration, but our results show it is not severe. SKOLR's performance compares favorably with Koopa's.
> ## Non-overlapping patch tokenization
> We apply patch processing before Linear RNN branches. This reduces complexity to O(L/P) from timestamps (O(L)). Our efficiency results *do* use this for baselines (PatchTST, iTransformer, Koopa). It slightly improves baseline performance.
> ## Hyperparameter selection + sensitivity
> We will clearly specify the hyperparameter selection protocol in App.A.2. We selected the optimal configuration based on MSE for the validation split. Our setup strictly separates training, validation, and test sets, ensuring no information leakage.
> Tab. 4 (paper) analyzes how branch number N and dimension D impact performance. Response [Tab. 3](https://anonymous.4open.science/r/SKOLR-1D6F/Tab3_PatchLen.pdf) provides new results of sensitivity to patch length. SKOLR exhibits little sensitivity to patch length P. We do not tune this in our experiments; we used P = L/6 for all datasets.
> ## Revise Tab. 8
> We apologize for inconsistencies in Tab.8. We've verified all results against our original records and fixed the MASE metric issues for the Quarter and Others categories in the revised [Tab.8](https://anonymous.4open.science/r/SKOLR-1D6F/Tab8_ShortTerm.pdf).
> ## Relationship to [1] Orvieto et al. 2023
> Thank you for highlighting this paper. We will modify the introduction: "In this work, we consider time-series forecasting, and establish a connection between Koopman operator approximation and linear RNNs, building on the observation made by [1]. We make a more explicit connection and devise an architecture that is a more direct match."
>
> In Related Work, we will add:
> "Orieto et al. made the observation, based on a qualitative discussion, that the Koopman operator representation of a dynamical system can be implemented by combining a wide MLP and a linear RNN. We provide a more explicit connection, providing equations to show a direct analogy between a structured approximation of a Koopman operator and an architecture comprised of an MLP encoder plus a linear RNN. Although [1] observe this connection, their studied architecture stacks linear recurrent units interspersed with non-linear activations or MLPs. While excellent for long-range reasoning tasks, this departs from the architecture in their App. E. By contrast, our developed architecture does consist of (multiple branches) of an MLP encoder, a single-layer linear RNN, and an MLP decoder. It thus adheres exactly to the demonstrated analogy between Eq. (5) and (8) of our paper. Whereas our focus is time series forecasting, [1] target long-range reasoning. Although it is possible to convert their architecture to address forecasting, performance suffers because it is not the design goal."
>
> We recognize the importance of acknowledging [1], but we don't believe that its existence significantly diminishes our contribution. The connection observed by Orieto et al. is qualitative; there are no supporting equations. In contrast, we develop an explicit connection by expressing a linear RNN in (5) and a structured Koopman operator approximation in (8). This explicit connection adds value beyond the qualitative insights.
> ## Presentation - Sec. 3
> We apologize for notational confusion and will correct it. There are strong motivations for the frequency decomposition design choice. In classical time-frequency analysis, the value of adaptation to different frequencies has long been recognized. Wavelet analysis applies different filters at different frequency scales. In more recent forecasting literature, frequency decomposition has been shown to be highly effective in TimesNet (Wu 2023), Koopa (Liu 2023), and MTST (Zhang 2024). Low-frequency dynamics may be considerably different from high-frequency dynamics and are more easily learnt after disentanglement. We are motivated to allow for observable functions to be learned both in frequency and time, with explicit consideration of the frequency aspect.
>
> We will modify Sec.3 to describe the whole pipeline for a single branch, and then describe the multiple-branch case.
> ## Questions: Non-linear systems; Fig. label
> Tab. 1 signals are real system measurements, e.g., Electricity Transformer Temperature. We do not know the exact dynamics. The 4 systems in Sec.4.2 are commonly-studied, synthetic non-linear systems. These allow us to study a setting where it is important to model non-linear dynamics. We will write "Synthetic Non-linear Systems" to stress the synthetic nature.
>
> The left figure of Fig.3 corresponds to (a) and the right to (b). We will add clear labels.

---

> > ### Comment · Reviewer_kXAS · 2025-04-03
> >
> > Thank you for providing the clarifications. The following is my revised understanding of the paper:
> >
> > - The paper proposes a new linear RNN for time-series forecasting. The key components proposed are block-diagonal state-transition matrix and learnable frequency-domain gating. The former is inspired by Koopman operator theory for deterministic nonlinear dynamical systems and the EDMD algorithm. The latter is inspired by existing works in time-series modeling leveraging frequency-domain analysis components.
> >
> > - Empirically, the proposed method overall shows a competitive performance compared to the state-of-the-art Koopa, and at the same time has a good time and space computational efficiency.
> >
> > I would like to see the other reviewers' comments on the author response before deciding my final score.

---

> > > ### Author Response · Authors · 2025-04-09
> > >
> > > Thank you for your thoughtful feedback and accurate summary of our paper's key contributions and empirical findings. We greatly appreciate your careful consideration of our clarifications and your concise articulation of our work on the linear RNN with block-diagonal state-transition matrix and learnable frequency-domain gating.
> > >
> > > Regarding the review process:
> > >
> > > > Reviewer hUoy has expressed positive willingness to consider raising their score after we addressed their specific questions. This was mentioned in the "Other Comments or Suggestions" section. We have now provided a response to each question.
> > >
> > > > Reviewer fgpm has noted "more confidence in being willing to support this paper" following our detailed responses, which is encouraging.
> > >
> > > > Reviewer qExE has not yet responded to the clarifying questions we posed regarding several of their claims in their review.
> > >
> > > We thank you and all reviewers for your positive engagement throughout this process. We understand your desire to review other reviewers' comments before finalizing your score, and we respect this thoughtful approach to the evaluation process. We're pleased that our responses have helped address your questions, and we remain available should you need any further clarification as you review the comments from other reviewers.

---

### Decision · Program_Chairs · 2025-05-01

**Decision:**

Accept (poster)

**Comment:**

the paper motivates the architecture, shows strong forecasting results across 8 real-world benchmarks and 4 nonlinear dynamical systems, and lower parameter/time cost compared to Koopa and Transformer baselines. The author's rebuttal addressed reviewers' main concerns (frequency-band “specialization” claim, statistical significance, efficiency analysis, additional ablations).

Reviewer qBxE remains opposed, citing inadequate formalism in the Koopman–RNN equivalence and missing assumptions/notation. The rebuttal provided a line-by-line response, clarifying assumptions, adding a proposition-proof outline, expanded ablations, and variance statistics which I think confirms validity of the results.

the qualitative link between Koopman operators and linear RNNs was hinted in prior work—but SKOLR turns this insight into a compact, easily parallelized architecture that attains consistently top or second-best accuracy with ~½ the memory of current SOTA. Given (i) broad interest in lightweight sequence models, (ii) solid empirical evidence with low variance, and (iii) responsiveness to reviewer feedback, the paper meets the bar for the conference.